# Text-Image Dual Consistency-Guided OOD Detection with Pretrained Vision-Language Models

## Abstract

The advent of vision-language models (VLMs) such as CLIP has significantly advanced the development of zero-shot out-of-distribution (OOD) detection. Recent research has largely focused on enhancing the textual label space to improve OOD detection performance. However, these efforts often neglect the valuable information inherent in the image domain. As a result, visual feature similarities within in-distribution (ID) data remain underutilized, limiting the OOD detection capabilities of VLMs. To address this limitation, we propose a novel approach, DualCnst, based on text-image dual consistency. Our method evaluates test samples by jointly considering their semantic similarity to textual labels and their visual similarity to synthesized images generated from the textual label set using a text-to-image generative model. By integrating textual and visual information, this approach establishes a unified OOD scoring framework. Furthermore, this framework is fully compatible with existing methods, such as NegLabel, which focus on enriching the textual label space. Extensive experiments demonstrate that DualCnst achieves state-of-the-art performance across a range of OOD detection benchmarks while exhibiting robust generalization across diverse VLM architectures.

## 1. Introduction

Out-of-Distribution (OOD) detection refers to identifying whether input data lies outside the predefined distribution of a machine learning model during inference (Hendrycks & Gimpel, 2017). Its primary goal is to prevent models from making erroneous predictions when confronted with novel or anomalous samples that deviate from the training data distribution. This capability is particularly critical in high-stakes applications, such as medical imaging (Shen et al., 2017; Wang et al., 2021; Kollias et al., 2024) and autonomous driving (Gao et al., 2021; Henriksson et al., 2023; Zhao et al., 2024), where undetected OOD samples can lead to misdiagnoses or hazardous situations.

Traditional visual OOD detection methods primarily rely on features extracted from the image domain, often neglecting the rich semantic information contained in textual labels. Recent advancements in large-scale vision-language models (VLMs) have shifted the focus toward leveraging multimodal information from both test images and textual labels to enhance OOD detection accuracy. For example, MCM (Ming et al., 2022) utilizes VLMs like CLIP (Radford et al., 2021) to compute semantic similarity between test images and in-distribution (ID) textual labels. Images with high similarity are classified as ID, while those with low similarity are deemed OOD. This approach enables zero-shot OOD detection without requiring additional training.

However, solely relying on semantic similarity has inherent limitations. Some challenging OOD samples, particularly those near the overlap of ID and OOD score distributions (Figure 1(b)), may share semantic features resembling those of ID labels, making them difficult to detect through semantic matching alone. Interestingly, such samples often remain visually distinguishable. For example, as shown in Figure 1(a), *wild horses and zebras are semantically similar but visually distinct due to the zebras' unique striped patterns.* This observation leads to the following question:

*Can incorporating visual similarity between test data and ID/OOD data improve detection accuracy for these challenging samples?*

Figure 1(c) confirms this hypothesis. By incorporating the actual visual features of ID data, challenging OOD samples become more distinguishable. However, real-world applications often face restricted access to ID visual features. To address this limitation, we propose synthesizing ID visual information using text-to-image generative models. These models generate image data for ID classes directly from textual prompts, without relying on any actual ID images, as illustrated in Figure 1(d). Additionally, we introduce synthesized OOD visual information, such as generating OOD images from negative labels (e.g., via NegLabel (Jiang et al., 2024)), to further enhance discriminative power.

To operationalize this idea, we propose DualCnst, a novel approach leveraging the dual consistency between test data, textual labels, and synthesized image labels. Technically, we develop a scoring function that evaluates the semantic similarity between test images and textual labels, while

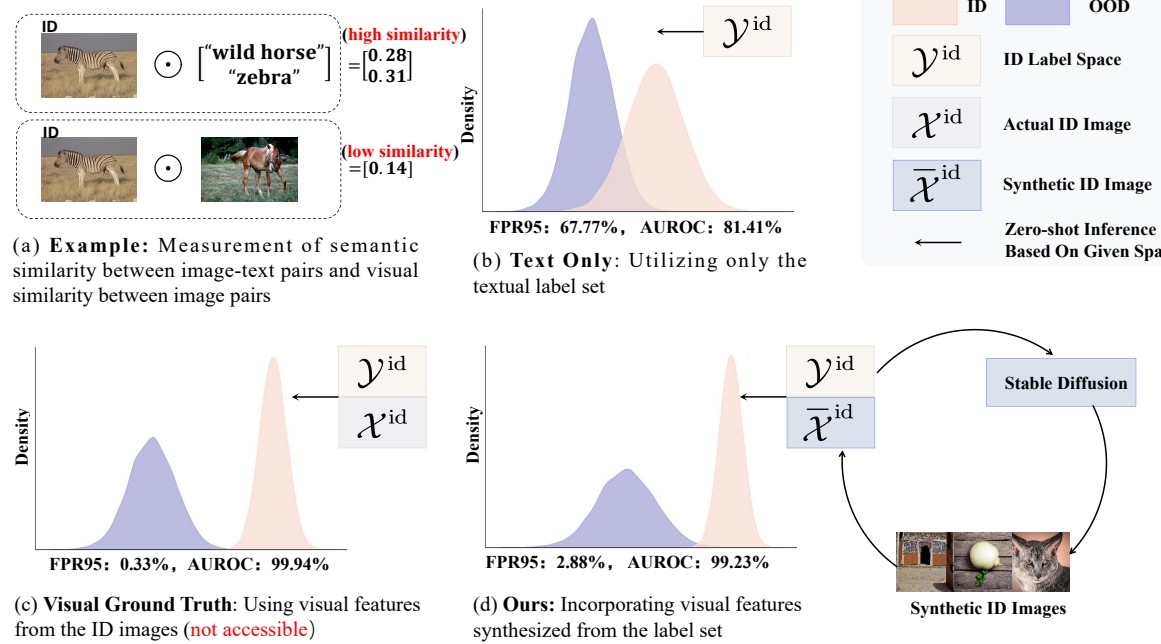

Figure 1: Comparison of zero-shot OOD detection score distribution. (a) The positive impact of visual similarity in detecting challenging OOD samples. Compared to the model using (b) only the textual label set, (c) incorporating visual features from the ID images significantly improves OOD detection performance. (d) Furthermore, by integrating visual features synthesized from the label set, OOD detection results can be substantially enhanced, even without utilizing the actual visual features from the ID images. Cifar100 (Krizhevsky et al., 2009) is used as the ID class, and iNaturalist (Van Horn et al., 2018) as the OOD class.

simultaneously measuring the visual similarity between the test data and synthesized ID/OOD image labels.

The proposed ID/OOD visual synthesis framework offers substantial performance improvements and several key advantages: (1) *Data-Agnostic*: It does not require actual visual information from ID/OOD data. (2) *Zero-Shot*: It supports diverse task-specific ID datasets using a single pretrained model. (3) *Scalability and Flexibility*: The visual similarity measure operates as a lightweight, plug-and-play module that can be seamlessly integrated into existing semantic similarity-based methods, making it adaptable across various datasets and applications.

Our contributions can be summarized as follows:

• A novel perspective is proposed, integrating visual feature similarity to address the limitations of relying solely on semantic features in distinguishing challenging OOD samples (Section 3).

• The DualCnst framework, a novel approach for zero-shot OOD detection, is introduced. It simultaneously evaluates the semantic similarity between test images and textual labels, while also leveraging synthesized ID/OOD image labels to assess the visual similarity between the test data and these synthesized labels (Section 3).

• The proposed DualCnst demonstrates superior performance, significantly outperforming existing methods. DualCnst

achieves improvements of 2.35%, 3.9%, 9.9% on the ImageNet-1K far OOD, near OOD, and Robust OOD detection tasks, respectively, in terms of FPR95 (Section 4).

## 2. Preliminaries

**CLIP and Zero-shot OOD Detection:** CLIP (Radford et al., 2021) is a multimodal pre-trained model designed to align visual and textual modalities within a shared embedding space. Trained on large-scale image-text datasets using contrastive learning, CLIP consists of an image encoder and a text encoder that generate embeddings for images and text, respectively. By computing cosine similarity between these embeddings, the model performs similarity-based matching. A key strength of CLIP is its remarkable zero-shot capability: trained on diverse and extensive image-text pairs, it can be directly applied to various vision tasks—including image classification (Conde & Turgutlu, 2021; Fu et al., 2022; Abdelfattah et al., 2023; Peng et al., 2023), object detection (Teng et al., 2021; Lin & Gong, 2023; Liu et al., 2024), semantic segmentation (Liang et al., 2023; Zhou et al., 2023; Wysoczańska et al., 2024), and OOD detection—without requiring additional labeled data or fine-tuning.

For zero-shot OOD detection, CLIP determines whether an input image belongs to one of the known categories or represents an OOD sample. This is achieved by comparing the image's visual features with the semantic representa-

Figure 2: The framework of the proposed DualCnst is outlined as follows. Given a set of ID class labels $\mathcal{Y}^{\mathrm{id}}$, we first leverage NegLabel (Jiang et al., 2024)) to generate OOD labels $\mathcal{Y}^{\mathrm{ood}}$. These class labels are then input into Stable Diffusion (Rombach et al., 2022) to synthesize both ID and OOD images. Subsequently, both the ID/OOD class labels and the synthesized images are fed into the text and image encoders to construct the textual and image classifiers. During the testing phase, given an input image, its visual features are extracted using the image encoder, and the semantic similarity with the class labels is computed, along with the visual similarity to the synthesized images. Finally, the OOD score is derived by scaling and coupling these similarities using the proposed detection score function $S_{\mathrm{DualCnst}}$.

tions of known class labels encoded as text. Images with low similarity to all known labels are identified as OOD samples. This zero-shot paradigm offers high flexibility, allowing CLIP to generalize across diverse domains without retraining, making it a powerful tool for OOD detection in real-world applications.

**Stable Diffusion:** Stable Diffusion is a generative model based on Latent Diffusion Models (LDMs) (Rombach et al., 2022), designed for efficient text-to-image synthesis. Unlike conventional diffusion models that operate in pixel space, Stable Diffusion performs the diffusion process in a lower-dimensional latent space, significantly enhancing computational efficiency and scalability. The model employs a pre-trained Variational Autoencoder (VAE) (Kingma, 2013) to encode high-resolution images into a compact latent representation, which serves as the input for the diffusion process. Within this latent space, a U-Net-based (Ronneberger et al., 2015) denoising network executes both forward and reverse diffusion: in the forward process, noise is gradually added to the latent representation until it converges to a Gaussian distribution, while in the reverse process, the model learns to iteratively denoise the latent representation, reconstructing it into the original data distribution.

To enhance the fidelity and semantic alignment of generated images, Stable Diffusion incorporates CLIP as a guidance mechanism during the reverse diffusion process. CLIP provides a similarity-based gradient signal that directs the latent representation toward alignment with the textual prompt, ensuring that the generated images faithfully capture both the semantic intent and fine-grained details. This method builds on previous CLIP-guided generative models (Galatolo et al., 2021; Desai et al., 2021; Qiao et al., 2022; Song et al., 2021), which utilize multimodal representations to improve the coherence and expressiveness of generated content. By leveraging CLIP's semantic understanding, Stable Diffusion generates visually coherent and contextually relevant images, even for abstract or complex prompts. This significantly broadens the model's applicability in text-to-image synthesis (Nichol et al., 2021).

## 3. Text-Image Dual Consistency-Guided OOD Detection

In this paper, a novel approach is proposed to enhance zero-shot OOD detection performance by leveraging text-image dual consistency. Specifically, the method is divided into two stages: (i) Synthesis Stage: To evaluate the visual sim-

ilarity of test samples with ID and OOD images, a text-to-image generative model, Stable Diffusion, is employed to synthesize image labels from the combined label space, $\mathcal{Y}^{\text{id}} \cup \mathcal{Y}^{\text{ood}}$. (ii) Testing Stage: To integrate textual and visual information, a novel score function is proposed. This function simultaneously evaluates the semantic similarity between test images and textual labels and measures the visual similarity between test samples and the synthesized ID/OOD image labels. The overall framework of the proposed method is illustrated in Figure 2.

### 3.1. Synthesize Images from the Label Space

To broaden the scope of visual information, NegLabel (Jiang et al., 2024) is employed to identify potential OOD labels, which serve as prompts for an image generator. These prompts guide the generation of semantically consistent visual representations for OOD images. The label space is defined as $\mathcal{Y}^{\text{id}} \cup \mathcal{Y}^{\text{ood}} = y_1, y_2, \ldots, y_K, y_{K+1}, \ldots, y_{K+M}$, where $K$ denotes the number of ID labels and $M$ denotes the number of OOD labels.

To ensure semantic alignment between textual descriptions and generated images, the diffusion model's capacity for aligning textual and visual representations is utilized. For each label, a consistent text prompt, such as "A photo of a <label>," is constructed. These prompts are input into the diffusion model to generate synthetic images semantically aligned with the combined label space $\mathcal{Y}^{\text{id}} \cup \mathcal{Y}^{\text{ood}}$. This process enriches visual information and addresses the limitations of relying solely on semantic information for image-text alignment.

The generated images are represented as $\bar{\mathcal{X}} = \{\bar{\mathbf{x}}_i\}$, where each $\bar{\mathbf{x}}_i$ corresponds to a unique synthetic image associated with a specific label. These images not only capture the known ID data distributions but also simulate visual representations of OOD categories. By integrating this diverse set of synthetic images into the OOD detection process, the proposed method enhances the model's ability to differentiate ID from OOD instances. This is achieved by leveraging visual distinctions between ID and OOD images, leading to more accurate identification and rejection of OOD samples.

### 3.2. Integrate Textual and Visual Metrics for OOD Detection

We calculate the visual similarity between the test sample $\mathbf{x}$ and the synthesized image set $\bar{\mathcal{X}}$, as well as the semantic similarity with the label set $\mathcal{Y}$, in the feature space encoded by CLIP's text encoder $\mathcal{T}(\cdot)$ and image encoder $\mathcal{I}(\cdot)$.

**Image-to-Image Similarity.** In particular, both low-level and high-level visual features are incorporated. We extract features from intermediate layers and the final output layer of the image encoder to calculate cosine similarity between the test sample and the synthetic images at multiple levels of representation. Distinct weights are assigned to each layer

to balance their contributions. For instance, using ViT-B/16 as the visual encoder, we select the third, sixth, ninth, and final semantic layers to compute cosine similarity between the test image and each synthetic image. A weight of $0.25$ is assigned to the similarity score from each layer, and the overall visual similarity is calculated as the weighted sum of these scores.

The visual similarity between the input image $\mathbf{x}$ and the synthesized image set $\bar{\mathcal{X}}$ is defined as:

$$s_{i,\text{img}}^{(l)}(\mathbf{x}) = \frac{\mathcal{I}^{(l)}(\mathbf{x}) \cdot \mathcal{I}^{(l)}(\bar{\mathbf{x}}_i)}{\|\mathcal{I}^{(l)}(\mathbf{x})\| \cdot \|\mathcal{I}^{(l)}(\bar{\mathbf{x}}_i)\|}; \quad \bar{\mathbf{x}}_i \in \bar{\mathcal{X}}. \quad (1)$$

where $\mathcal{I}^{(l)}(\mathbf{x})$ represents the feature embedding at layer $l$. The final similarity score $s_{i,\text{img}}(\mathbf{x})$ is obtained by summing the weighted similarity scores across all layers:

$$s_{i,\text{img}}(\mathbf{x}) = \sum_{l=1}^{L} w_l \cdot s_{i,\text{img}}^{(l)}(\mathbf{x}), \quad (2)$$

where $w_l$ represents the weight assigned to layer $l$ and is defined as:

$$w_l = \begin{cases} r, & l < L \\ 1 - r \cdot (L-1), & l = L \end{cases},$$

where $L$ denotes the total number of layers in the visual encoder, and $r$ is the weight factor applied to intermediate layers, ensuring a balanced contribution across all layers.

**Image-to-Text Similarity.** The semantic similarity between the test image $\mathbf{x}$ and the combined label space $\mathcal{Y}^{\text{id}} \cup \mathcal{Y}^{\text{ood}}$ is computed as:

$$s_{i,\text{text}}(\mathbf{x}) = \frac{\mathcal{I}(\mathbf{x}) \cdot \mathcal{T}(\mathbf{t}_i)}{\|\mathcal{I}(\mathbf{x})\| \cdot \|\mathcal{T}(\mathbf{t}_i)\|}. \quad (3)$$

where $\mathbf{t}_i = \text{prompt} < y_i >$ and $y_i \in \mathcal{Y}^{\text{id}} \cup \mathcal{Y}^{\text{ood}}$, and $\mathbf{t}_i$ represents the textual description of the label $y_i$, using a prompt format such as "A photo of a <label>."

**Fusion of Similarity Scores.** To fully utilize both image-to-image and image-to-text similarity information, we compute a fused similarity score using a weighted sum-softmax method:

$$S_{\text{DualCnst}}(\mathbf{x}) = \sum_{i=1}^{K} \frac{\exp(\tilde{s}_i(\mathbf{x}))}{\sum_{j=1}^{K+M} \exp(\tilde{s}_j(\mathbf{x}))}, \quad (4)$$

where the fused similarity score $\tilde{s}_i(\mathbf{x})$ is defined as:

$$\tilde{s}_i(\mathbf{x}) = \alpha \cdot s_{i,\text{img}}(\mathbf{x}) + (1-\alpha) \cdot s_{i,\text{text}}(\mathbf{x}), \quad (5)$$

where $\alpha$ is a fusion hyperparameter that balances the contributions of image-to-image and image-to-text similarities. Details on the choice of $\alpha$ are provided in Appendix B.4.

**Algorithm 1** Zero-shot OOD detection with text-image dual consistency

---

1: **Input:** ID class labels $\mathcal{Y}^{\text{id}}$, test sample $\mathbf{x}$, text encoder $\mathcal{T}$, image encoder $\mathcal{I}$, Stable Diffusion (SD), NegLabel, fusion coefficient $\alpha$, layer weight $w$, threshold $\lambda$;
   **Synthesis stage:**
   // Synthesize OOD class labels
2: Given $\mathcal{Y}^{\text{id}}$, $\mathcal{Y}^{\text{ood}} = \text{NegLabel}(\mathcal{Y}^{\text{id}})$;
   // Synthesize ID/OOD image labels
3: Given $\mathcal{Y}^{\text{id}} \cup \mathcal{Y}^{\text{ood}}$, $\bar{\mathcal{X}} = \text{SD}(\text{prompt} < \mathcal{Y}^{\text{id}} \cup \mathcal{Y}^{\text{ood}} >)$;
   **Testing stage:**
   // Calculate image-to-image similarity
4: $s_{i,\text{img}}^{(l)}(\mathbf{x}) = \frac{\mathcal{I}^{(l)}(\mathbf{x}) \cdot \mathcal{I}^{(l)}(\bar{\mathbf{x}}_i)}{\|\mathcal{I}^{(l)}(\mathbf{x})\| \cdot \|\mathcal{I}^{(l)}(\bar{\mathbf{x}}_i)\|}$; $\bar{\mathbf{x}}_i \in \bar{\mathcal{X}}$;
5: $s_{i,\text{img}}(\mathbf{x}) = \sum_{l=1}^{L} w_l \cdot s_{i,\text{img}}^{(l)}(\mathbf{x})$;
   // Calculate image-to-text similarity
6: $\mathbf{t}_i = \text{prompt} < y_i >$; $y_i \in \mathcal{Y}^{\text{id}} \cup \mathcal{Y}^{\text{ood}}$;
7: $s_{i,\text{text}}(\mathbf{x}) = \frac{\mathcal{I}(\mathbf{x}) \cdot \mathcal{T}(\mathbf{t}_i)}{\|\mathcal{I}(\mathbf{x})\| \cdot \|\mathcal{T}(\mathbf{t}_i)\|}$;
   // Integrate text and visual information
8: $\tilde{s}_i(\mathbf{x}) = \alpha \cdot s_{i,\text{img}}(\mathbf{x}) + (1 - \alpha) \cdot s_{i,\text{text}}(\mathbf{x})$;
   // Calculate OOD detection score
9: $S_{\text{DualCnst}}(\mathbf{x}) = \sum_{i=1}^{K} \frac{\exp(\tilde{s}_i(\mathbf{x}))}{\sum_{j=1}^{K+M} \exp(\tilde{s}_j(\mathbf{x}))}$;
10: **Output:** ID if $S_{\text{DualCnst}}(\mathbf{x}) > \lambda$, else **OOD**.

---

**OOD Detection Framework.** Based on $S_{\text{DualCnst}}(\mathbf{x})$, the OOD detector $G_\lambda(\mathbf{x}; \mathcal{Y}^{\text{id}} \cup \mathcal{Y}^{\text{ood}}, \mathcal{T}, \mathcal{I})$ is defined as a binary classification function:

$$G_\lambda(\mathbf{x}; \mathcal{Y}^{\text{id}} \cup \mathcal{Y}^{\text{ood}}, \bar{\mathcal{X}}, \mathcal{T}, \mathcal{I}) = \begin{cases} \text{ID} & S_{\text{DualCnst}}(\mathbf{x}) \geq \lambda \\ \text{OOD} & S_{\text{DualCnst}}(\mathbf{x}) < \lambda \end{cases}, \quad (6)$$

where $\lambda$ is a threshold selected such that a high fraction of ID samples (typically 95%) exceed this value. See Algorithm 1 for the complete zero-shot OOD detection procedure.

## 4. Experiments

### 4.1. Experiment Setup

**Datasets and Benchmarks.** For our experiments, we use ImageNet-1k (Deng et al., 2009) as the primary ID dataset. OOD datasets include iNaturalist (Van Horn et al., 2018), SUN (Xiao et al., 2010), Places (Zhou et al., 2017), and Textures (Cimpoi et al., 2014), which cover a wide variety of scenes and semantic categories. We also adopt the experimental setup from MCM (Ming et al., 2022), which leverages subsets of ImageNet-1k to evaluate our method. Specifically, ImageNet-10 and ImageNet-20 are alternately used as ID and OOD datasets. Furthermore, we extend our evaluation to more generalized ImageNet variants, including ImageNet-R (Hendrycks et al., 2021a).

**Implementation Details.** Our framework is built upon CLIP (Radford et al., 2021) as the core model. Unless otherwise noted, we utilize the ViT-B/16 architecture as the image encoder and a Masked Self-Attention Transformer (Vaswani

et al., 2017) as the text encoder. For image generation, we employ the Stable Diffusion. We set $\alpha = 0.1$ and $w = 0.1$, and provide ablation experiments. Further details can be found in Appendix B. To improve inference efficiency, all synthetic images are pre-generated before the evaluation phase, eliminating the need for additional computational overhead during testing. Further details in Appendix C.3.

For evaluation, we use two primary metrics: (1) **FPR95**: The false positive rate (FPR) at a true positive rate (TPR) of 95% for ID data. (2) **AUROC**: The area under the receiver operating characteristic curve. Additionally, we report the results in terms of **AUPR** in Appendix C.2.

**Baseline Methods.** We benchmark our method against several state-of-the-art zero-shot OOD detection approaches, including Mahalanobis Distance (Lee et al., 2018), Energy Score (Liu et al., 2020), ZOC (Esmaeilpour et al., 2022), MCM (Ming et al., 2022), and NegLabel (Jiang et al., 2024). Additionally, we compare our approach with OOD detection models that have been trained or fine-tuned using ID data, such as MOS (Huang & Li, 2021), MSP (Hendrycks & Gimpel, 2017), CLIPN (Wang et al., 2023), VOS (Du et al., 2022), and NPOS (Tao et al., 2023).

### 4.2. Main Results

**Performance Comparison of ImageNet-1k on Far OOD Detection.** We compare our method with several existing OOD detection approaches, as shown in Table 1. These include zero-shot OOD detection methods such as MCM, EOE, and NegLabel, as well as traditional methods that re-implement CLIP fine-tuned on ImageNet-1k. Our approach achieves the best performance on ImageNet-1k. Compared to the current best-performing method, NegLabel, our method reduces the average FPR95 by 1.75% and improves the average AUROC by 0.14%. Moreover, it outperforms NegLabel on all OOD datasets.

The limited improvement observed on the Textures dataset is primarily attributed to the inherent challenges posed by this dataset. We believe this is due to the relatively constrained capabilities of the Stable Diffusion model, as well as insufficiently detailed prompt descriptions used to generate synthetic images. These factors contribute to synthetic images that exhibit discrepancies in both semantic alignment and pixel-level representation with the Textures dataset. Nevertheless, our method consistently achieves state-of-the-art results, demonstrating its robustness and effectiveness even in the face of such challenges.

**Performance Comparison of Different ID Datasets on Far OOD Detection.** Table 2 presents the performance of our method across seven distinct ID datasets: CUB-200-2011 (Wah et al., 2011), Stanford-Cars (Krause et al., 2013), Food-101 (Bossard et al., 2014), Oxford-IIIT Pet (Parkhi et al., 2012), ImageNet-10, ImageNet-20, and ImageNet-

Table 1: Performance Comparison of ImageNet-1k on Far OOD Detection. The **bold** indicates the best performance on each dataset, and the gray indicates methods requiring an additional massive auxiliary dataset.

| Method | iNaturalist | | SUN | | Places | | Textures | | Average | |
|---|---|---|---|---|---|---|---|---|---|---|
| | FPR95↓ | AUROC↑ | FPR95↓ | AUROC↑ | FPR95↓ | AUROC↑ | FPR95↓ | AUROC↑ | FPR95↓ | AUROC↑ |
| MOS (BiT) (Huang & Li, 2021) | 9.28 | 98.15 | 40.63 | 92.01 | 49.54 | 89.06 | 60.43 | 81.23 | 39.97 | 90.11 |
| MSP (Hendrycks & Gimpel, 2017) | 40.89 | 88.63 | 65.81 | 81.14 | 67.90 | 80.14 | 64.96 | 78.16 | 59.89 | 82.04 |
| CLIPN (Wang et al., 2023) | 19.13 | 96.20 | 25.69 | 94.18 | 32.14 | **92.26** | 44.60 | 88.93 | 30.39 | 92.89 |
| VOS (Du et al., 2022) | 28.99 | 94.62 | 36.88 | 92.57 | 38.39 | 91.23 | 61.02 | 86.33 | 41.32 | 91.19 |
| NPOS (Tao et al., 2023) | 16.58 | 96.19 | 43.77 | 90.44 | 45.27 | 89.44 | 46.12 | 88.80 | 37.93 | 91.22 |
| Mahalanobis (Lee et al., 2018) | 99.33 | 55.89 | 99.41 | 59.94 | 98.54 | 65.96 | 98.46 | 64.23 | 98.94 | 61.50 |
| Energy (Liu et al., 2020) | 81.08 | 85.09 | 79.02 | 84.24 | 75.08 | 83.38 | 93.65 | 65.56 | 82.21 | 79.57 |
| ZOC (Esmaeilpour et al., 2022) | 87.30 | 86.09 | 81.51 | 81.20 | 73.06 | 83.39 | 98.90 | 76.46 | 85.19 | 81.79 |
| MCM (Ming et al., 2022) | 30.91 | 94.61 | 37.59 | 92.57 | 44.69 | 89.77 | 57.77 | 86.11 | 42.74 | 90.77 |
| NegLabel (Jiang et al., 2024) | 1.91 | 99.49 | 20.53 | 95.49 | 35.59 | 91.64 | 43.56 | 90.22 | 25.40 | 94.21 |
| DualCnst | **0.99** | **99.69** | **17.60** | **95.89** | **31.63** | 91.73 | **42.00** | **90.32** | **23.05** | **94.41** |

Table 2: Performance Comparison of Different ID Datasets on Far OOD Detection. The **bold** indicates the best performance on each dataset.

| ID Dataset | Method | OOD Dataset | | | | | | | | Average | |
|---|---|---|---|---|---|---|---|---|---|---|---|
| | | iNaturalist | | SUN | | Places | | Texture | | | |
| | | FPR95↓ | AUROC↑ | FPR95↓ | AUROC↑ | FPR95↓ | AUROC↑ | FPR95↓ | AUROC↑ | FPR95↓ | AUROC↑ |
| Stanford-Cars | MCM (Ming et al., 2022) | 0.05 | 99.77 | 0.02 | 99.95 | 0.24 | 99.89 | 0.02 | 99.96 | 0.08 | 99.89 |
| | NegLabel (Jiang et al., 2024) | 0.01 | 99.99 | 0.01 | 99.99 | 0.03 | 99.99 | 0.01 | 99.99 | 0.01 | 99.99 |
| | DualCnst | **0.00** | **100.00** | **0.00** | **100.00** | **0.03** | **99.99** | **0.00** | **100.00** | **0.01** | **100.00** |
| CUB-200 | MCM (Ming et al., 2022) | 9.83 | 98.24 | 4.93 | 99.10 | 6.65 | 98.57 | 6.97 | 98.75 | 7.09 | 98.66 |
| | NegLabel (Jiang et al., 2024) | 0.18 | 99.96 | 0.02 | 99.99 | **0.33** | **99.90** | 0.01 | 99.99 | 0.13 | 99.96 |
| | DualCnst | **0.12** | **99.98** | **0.02** | **99.99** | 0.38 | 99.89 | **0.00** | **100.00** | 0.13 | **99.96** |
| Oxford-Pet | MCM (Ming et al., 2022) | 2.85 | 99.38 | 1.06 | 99.73 | 2.11 | 99.56 | 0.80 | 99.81 | 1.70 | 99.62 |
| | NegLabel (Jiang et al., 2024) | 0.01 | 99.99 | 0.02 | 99.99 | 0.17 | 99.96 | 0.11 | 99.97 | 0.07 | 99.98 |
| | DualCnst | **0.00** | **100.00** | **0.00** | **100.00** | **0.15** | **99.97** | **0.09** | **99.98** | **0.06** | **99.99** |
| Food-101 | MCM (Ming et al., 2022) | 0.64 | 99.78 | 0.90 | 99.75 | 1.86 | 99.58 | 4.04 | 98.62 | 1.86 | 99.43 |
| | NegLabel (Jiang et al., 2024) | 0.01 | 99.99 | 0.01 | 99.99 | 0.01 | 99.99 | 1.61 | **99.60** | 0.40 | **99.90** |
| | DualCnst | **0.00** | **100.00** | **0.00** | **100.00** | **0.01** | **100.00** | **1.52** | 99.57 | **0.38** | 99.89 |
| ImageNet-10 | MCM (Ming et al., 2022) | 0.12 | 99.80 | 0.29 | 99.79 | 0.88 | 99.62 | 0.04 | 99.90 | 0.33 | 99.78 |
| | NegLabel (Jiang et al., 2024) | 0.02 | 99.83 | 0.20 | 99.88 | 0.71 | 99.75 | 0.02 | 99.94 | 0.24 | 99.85 |
| | DualCnst | **0.01** | **99.97** | **0.09** | **99.93** | **0.57** | **99.75** | **0.02** | **99.96** | **0.17** | **99.90** |
| ImageNet-20 | MCM (Ming et al., 2022) | 1.02 | 99.66 | 2.55 | 99.50 | 4.40 | 99.11 | 2.43 | 99.03 | 2.60 | 99.32 |
| | NegLabel (Jiang et al., 2024) | 0.15 | 99.95 | 1.93 | 99.51 | 4.40 | 98.97 | 2.41 | 99.11 | 2.22 | 99.39 |
| | DualCnst | **0.13** | **99.97** | **1.22** | **99.66** | **3.66** | **99.13** | **2.18** | **99.17** | **1.80** | **99.48** |
| ImageNet-100 | MCM (Ming et al., 2022) | 18.13 | 96.77 | 36.45 | 94.54 | 34.52 | 94.36 | 41.22 | 92.25 | 32.58 | 94.48 |
| | NegLabel (Jiang et al., 2024) | 0.53 | 99.87 | 9.91 | 98.12 | 20.26 | 96.18 | 25.50 | 95.27 | 14.05 | 97.36 |
| | DualCnst | **0.41** | **99.90** | **8.68** | **98.34** | **18.72** | **96.43** | **23.51** | **95.72** | **12.83** | **97.60** |

100. For each dataset, we set $\alpha = 0.1$, select the 3rd, 6th, and 9th layers of the visual encoder, and assign a weight of $w = 0.15$. Our method demonstrates robust performance across various datasets, underscoring its generalizability.

**Performance Comparison of ImageNet Subsets on Near OOD Detection.** Table 3 presents the experimental results with ImageNet-10 and ImageNet-20 used interchangeably as ID and OOD datasets. When ImageNet-10 was the ID dataset and ImageNet-20 the OOD dataset, our method achieved a 2.4% reduction in FPR95 and a 0.1% increase in AUROC compared to NegLabel. Similarly, when ImageNet-20 was the ID dataset and ImageNet-10 the OOD dataset, our method reduced FPR95 by 5.4% and improved AUROC by 0.4%. The subset division and ID label configurations follow the settings in MCM (Ming et al., 2022). For a fair comparison, we reproduced the results of NegLabel and MCM under these conditions.

**Performance Comparison on Robust OOD Detection.** To assess the generalization ability of our method under domain shifts, we conducted experiments using the ImageNet Domain Shift dataset, with ImageNet-R serving as the ID dataset. Table 4 presents the results based on CLIP-B/16 with $\alpha = 0.1$, selecting the 3rd, 6th, and 9th layers of the visual encoder, and assigning a weight of $w = 0.15$. Our method demonstrates stronger generalization performance compared to NegLabel. ImageNet-R (Hendrycks et al., 2021a) consists of 30,000 images spanning 200 ImageNet categories, with representations in diverse artistic styles, including art, cartoons, graffiti, embroidery, graphics, origami, paintings, patterns, plastic objects, plush objects, sculptures, sketches, tattoos, toys, and video game renditions.

### 4.3. Ablation Study

**Score Functions.** To demonstrate the superiority of the proposed OOD detection score $S_{\text{DualCnst}}$, we present the av-

Table 3: Performance Comparison of ImageNet Subsets on Near OOD Detection. The **bold** indicates the best performance on each dataset, and the `gray` indicates methods requiring an additional massive auxiliary dataset.

| Method | ID OOD | ImageNet-10 ImageNet-20 | | ImageNet-20 ImageNet-10 | | Average | |
|---|---|---|---|---|---|---|---|
| | | FPR95↓ | AUROC↑ | FPR95↓ | AUROC↑ | FPR95↓ | AUROC↑ |
| CLIPN (Wang et al., 2023) | | 7.80 | 98.07 | 13.67 | 97.47 | 10.74 | 97.77 |
| MaxLogit (Hendrycks & Gimpel, 2017) | | 9.70 | 98.09 | 14.00 | 97.81 | 11.85 | 97.95 |
| Energy (Liu et al., 2020) | | 10.30 | 97.94 | 16.40 | 97.37 | 13.35 | 97.66 |
| MCM (Ming et al., 2022) | | 5.00 | 98.71 | 17.40 | **97.87** | 11.20 | **98.29** |
| NegLabel (Jiang et al., 2024) | | 5.10 | 98.86 | 17.60 | 97.04 | 11.35 | 97.95 |
| DualCnst | | **2.20** | **98.96** | **12.20** | 97.44 | **7.45** | 98.20 |

Table 4: Robustness results on ImageNet-R dataset. The **black bold** indicates the best performance.

| Method | iNaturalist | | SUN | | OOD Dataset Places | | Texture | | Average | |
|---|---|---|---|---|---|---|---|---|---|---|
| | FPR95↓ | AUROC↑ | FPR95↓ | AUROC↑ | FPR95↓ | AUROC↑ | FPR95↓ | AUROC↑ | FPR95↓ | AUROC↑ |
| Energy (Liu et al., 2020) | 99.91 | 30.36 | 99.33 | 33.20 | 98.84 | 34.74 | 99.56 | 23.09 | 99.41 | 30.35 |
| MaxLogit (Hendrycks & Gimpel, 2017) | 86.53 | 81.58 | 82.11 | 81.48 | 78.16 | 79.86 | 91.24 | 69.45 | 84.51 | 78.09 |
| MCM (Ming et al., 2022) | 51.59 | 92.24 | 52.88 | 89.97 | 52.04 | 88.01 | 56.45 | 85.65 | 53.24 | 88.97 |
| NegLabel (Jiang et al., 2024) | 1.60 | 99.58 | 15.77 | 96.03 | 29.48 | 91.97 | 35.67 | 90.60 | 20.63 | 94.54 |
| DualCnst | **0.59** | **99.86** | **8.92** | **98.19** | **19.27** | **95.20** | **14.13** | **95.50** | **10.73** | **97.19** |

erage results on the ImageNet-1K dataset in Figure 3 (a), comparing it with other scoring functions: $S_{MAX}$, $S_{Energy}$, and $S_{MaxLogit}$. All these functions are specifically designed for the Dual Consistency approach. Please refer to Appendix B.6 for the specific forms and results on more datasets. Results show that our $S_{DualCnst}$ achieves the best OOD performance. This verifies the superiority and importance of the proposed OOD detection score.

**Different Layers of the Visual Encoder.** To explore the effectiveness of pixel-level features from different layers of the visual encoder, we sample various pixel layers and assign different weights, as shown in Figure 3 (b). Specifically, we experiment by selecting the (1st, 2nd, 3rd) layers, (4th, 5th, 6th) layers, (7th, 8th, 9th) layers, (9th, 10th, 11th) layers, and all pixel layers to combine with semantic layers. In Figure 3 (c), we further investigate the impact of different weight distributions for $w$ to identify the most suitable pixel-level feature weighting. For details on the selection of $w$, layers, and results, refer to Appendix B.3.

### 4.4. Further Analysis

**More Experimental Results**. We conducted experiments on the CIFAR-10/CIFAR-100 (Krizhevsky et al., 2009) benchmark to further validate our method. The details of the ImageNet-A (Hendrycks et al., 2021b) and ImageNet-V2 (Recht et al., 2019) generalization datasets are also provided in the AppendixA.2. Additionally, we explored the impact of randomness introduced by Stable Diffusion when generating synthetic images with different random seeds, as demonstrated in Table16 . The results show that the effect of Stable Diffusion's randomness on our method is negligible. It is important to note that we did not manually select the most favorable random seed for Stable Diffusion. Instead, we generated a 32-bit integer random seed by hashing the

combination of each class label and synthetic image index. Each synthetic image generated for a class using this seed exhibits substantial randomness, further demonstrating that our method is not influenced by the randomness of Stable Diffusion-generated images. We also conducted experiments with different CLIP visual encoders, and the results showed that stronger visual encoders, which capture more detailed information, are more beneficial to our method. For more details, please refer to Appendix B.1.

**Effectiveness of DualCnst.** Figure 4 shows the T-SNE (Van Der Maaten, 2014) visualization of the softmax outputs. We compare the results of NegLabel and DualCnst, using the ImageNet-10 dataset for ID and ImageNet-20 dataset for OOD. In this setup, there are several semantically similar pairs of ID and OOD categories, such as: horse (ID) vs. zebra (OOD), Swiss mountain dog (ID) vs. timberwolf (OOD), warplane (ID) vs. space shuttle (OOD), and garbage truck (ID) vs. steam locomotive (OOD). In the presence of such datasets, methods that expand the label space, like NegLabel, often struggle to find labels with a high overlap probability with true OOD labels, leading to suboptimal performance. As shown in (a) with the black bounding box, it is difficult to distinguish between ID and OOD samples, as they tend to interweave. DualCnst, however, addresses this issue by leveraging visual information to differentiate between ID and OOD samples. As demonstrated in (b), we incorporate visual information into NegLabel, allowing for better differentiation based on unique visual features inherent to ID and OOD samples, such as the stripes on a zebra or the ears and fur of a timberwolf. These observations indicate that DualCnst enables a significant improvement in the classifier's ability, making semantically similar ID and OOD samples more separable.

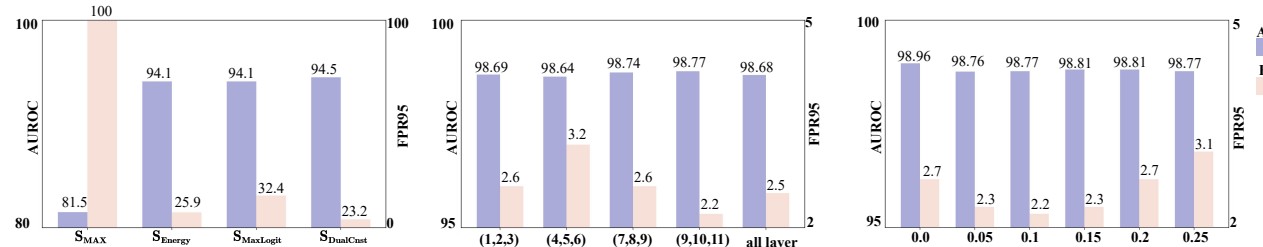

Figure 3: Ablation study on (a) score function, (b) Different Layers, and (c) Different Weight. ID dataset: ImageNet-10; OOD dataset: ImageNet-20.

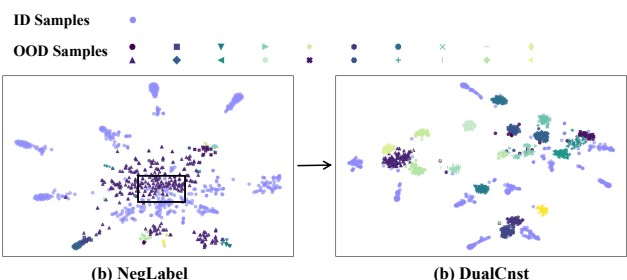

Figure 4: T-SNE visualizations obtained by the classifier output. ID set: ImageNet-10; OOD set: ImageNet-20. We use distinct colors to represent different OOD classes. Our DualCnst method achieves better separability between ID and OOD classes compared to NegLabel.

## 5. Related Works

**OOD Detection.** Early methods for OOD detection include classification-based approaches that rely on a well-trained ID classifier, such as MSP (Hendrycks & Gimpel, 2017). Density-based methods, such as likelihood ratios (Ren et al., 2019) and likelihood regret (Xiao et al., 2020), estimate the likelihood of data points to identify OOD samples. Reconstruction-based methods (Denouden et al., 2018; Zhou, 2022; Liu et al., 2023) leverage reconstruction errors from generative models, including VAEs and autoencoders, to detect OOD instances. Post-hoc methods, including ODIN (Liang et al., 2017) and energy-based scoring (Liu et al., 2020), enhance pre-trained models without modifying their parameters. More recently, multimodal vision-language models such as CLIP and its variants (Yuan et al., 2021) have enabled zero-shot OOD detection by leveraging text-image embeddings, marking a shift toward more versatile and scalable solutions.

**Zero-shot OOD Detection.** Recent advancements in zero-shot OOD detection take advantage of the powerful pre-training capabilities of models like CLIP, allowing for efficient OOD detection without the need for large external OOD labels. ZOC (Esmaeilpour et al., 2022) introduces a CLIP-based framework for zero-shot OOD detection, where potential OOD labels are generated for input instances using image captions, aligning images and text for zero-shot classification. MCM (Ming et al., 2022) performs OOD detection by utilizing scaled softmax values of the maximum

logits as confidence scores, but it relies solely on ID class labels and does not fully exploit open-world textual information. CLIPN (Wang et al., 2023) improves the model's ability to reject mismatched inputs by introducing learnable "negative" prompts and a dedicated "negative" text encoder. EOE (Cao et al., 2024) utilizes the expert knowledge and reasoning abilities of large language models (LLMs) to generate potential anomalies, enabling more effective OOD detection. NegLabel (Jiang et al., 2024) proposes a novel method that enhances the distinguishability between ID and OOD samples by mining potential OOD labels from a corpus. However, these methods do not fully consider the visual effectiveness of images. In contrast, DualCnst addresses this limitation by making semantically similar ID and OOD samples more distinguishable. Moreover, it can be seamlessly integrated into existing OOD frameworks.

**Stable Diffusion for OOD Detection.** Stable Diffusion has been explored for OOD detection in several studies. LMD (Liu et al., 2023) introduces a diffusion-based approach for image inpainting, where the input image is reconstructed, and the reconstruction error is used as an indicator for OOD detection. In contrast, DualCnst employs Stable Diffusion for image generation, offering a more efficient solution in open-world scenarios. Unlike LMD, DualCnst reduces the computational burden on the inference process, making it a more practical and scalable approach for OOD detection in dynamic environments.

## 6. Conclusion

In this paper, a novel perspective was introduced that incorporated visual metrics to improve detection accuracy for challenging OOD samples that were semantically similar to ID data. Building on this, the DualCnst framework was proposed as an innovative approach for zero-shot OOD detection. Specifically, test samples were evaluated by simultaneously analyzing their semantic similarity to textual labels and their visual similarity to synthesized images generated from the textual label set using a text-to-image generative model. Finally, extensive experiments validated the effectiveness of this perspective, demonstrating that Dual-Consistency achieved state-of-the-art performance across various OOD detection benchmarks.

## Impact Statement

This paper presents work whose goal is to advance the field of machine learning. There are many potential societal consequences of our work, none of which we feel must be specifically highlighted here.

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

# Appendix

## A. Further Experiments

### A.1. Robustness to Domain Shift

Table 5 presents an evaluation of DualCnst's robustness using the ImageNet-A (Hendrycks et al., 2021b) generalization dataset as the ID dataset, while iNaturalist (Van Horn et al., 2018), SUN (Xiao et al., 2010), Places (Zhou et al., 2017), and Textures (Cimpoi et al., 2014) serve as OOD datasets. We compare DualCnst against state-of-the-art methods. DualCnst outperforms NegLabel across all datasets, achieving an improvement of 2.09% in FPR95 and 0.25% in AUROC on average.

In Table 6, we further investigate the robustness of DualCnst under the same experimental setup using another generalization dataset, ImageNet-V2 (Recht et al., 2019). The experimental results demonstrate that our proposed method exhibits superior performance in handling domain shifts.

Table 5: Robustness results on ImageNet-A dataset. The ID class labels are the same as ImageNet. The **black bold** indicates the best performance.

| Method | OOD Dataset | | | | | | | | | |
|---|---|---|---|---|---|---|---|---|---|---|
| | iNaturalist | | SUN | | Places | | Texture | | Average | |
| | FPR95↓ | AUROC↑ | FPR95↓ | AUROC↑ | FPR95↓ | AUROC↑ | FPR95↓ | AUROC↑ | FPR95↓ | AUROC↑ |
| Energy | 99.48 | 50.03 | 95.01 | 58.83 | 93.52 | 60.86 | 97.46 | 42.18 | 96.37 | 52.97 |
| MaxLogit | 92.88 | 74.14 | 81.54 | 80.55 | 78.51 | 79.06 | 90.00 | 69.41 | 85.73 | 75.79 |
| MCM | 80.41 | 77.02 | 76.12 | 78.92 | 76.90 | 76.48 | 74.10 | 77.36 | 76.88 | 77.45 |
| NegLabel | 4.09 | 98.80 | 44.38 | 89.83 | 60.10 | 82.88 | 64.34 | 80.25 | 43.23 | 87.94 |
| DualCnst | **3.54** | **98.99** | **32.41** | **92.79** | **48.66** | **87.04** | **47.77** | **89.54** | **33.09** | **92.09** |

### A.2. Other OOD Detection Benchmarks

In Table 7, we present the performance evaluation results using CIFAR-10 and CIFAR-100 (Krizhevsky et al., 2009) as the ID datasets, along with four OOD datasets: iNaturalist (Van Horn et al., 2018), SUN (Xiao et al., 2010), Places (Zhou

Table 6: Robustness results on ImageNet-V2 dataset. The ID class labels are the same as ImageNet. The **black bold** indicates the best performance.

| Method | iNaturalist | | SUN | | Places | | Texture | | Average | |
|--------|---------|---------|---------|---------|---------|---------|---------|---------|---------|---------|
| | FPR95↓ | AUROC↑ | FPR95↓ | AUROC↑ | FPR95↓ | AUROC↑ | FPR95↓ | AUROC↑ | FPR95↓ | AUROC↑ |
| Energy | 99.85 | 32.93 | 99.12 | 34.45 | 98.02 | 39.51 | 99.57 | 21.52 | 99.14 | 32.10 |
| MaxLogit | 83.78 | 83.84 | 83.55 | 81.79 | 80.27 | 80.33 | 93.51 | 64.34 | 85.28 | 77.58 |
| MCM | 44.89 | 92.14 | 51.17 | 89.69 | 56.73 | 86.44 | 69.57 | 81.51 | 55.10 | 87.56 |
| NegLabel | 2.47 | 99.40 | 25.69 | 94.46 | 42.03 | 90.00 | **48.90** | **88.46** | 29.77 | 93.08 |
| DualCnst | **1.49** | **99.60** | **21.90** | **94.92** | **36.71** | **90.62** | 50.62 | 88.18 | **27.68** | **93.33** |

et al., 2017), and Textures (Cimpoi et al., 2014). Compared to the NegLabel method, our approach demonstrates significant performance gains. Specifically, on CIFAR-100, DualCnst achieves an average improvement of 23.59% in FPR95 and 9.34% in AUROC. On CIFAR-10, it yields improvements of 7.56% in FPR95 and 1.39% in AUROC. Although DualCnst does not achieve the best performance on CIFAR-10 individually, it outperforms existing methods in terms of overall average performance across both CIFAR-10 and CIFAR-100, highlighting its effectiveness in OOD detection across diverse datasets.

Additionally, in Table 8, we follow the fine-grained dataset setup proposed by EOE (Cao et al., 2024) and conduct experiments on CUB-200-2011 (Wah et al., 2011), STANFORD-CARS (Krause et al., 2013), Food-101 (Bossard et al., 2014), and Oxford-IIIT Pet (Parkhi et al., 2012).Under this experimental setting, the four datasets are randomly split into two equal subsets, with one serving as the ID dataset and the other as the OOD dataset. Since NegLabel identifies the most semantically distant candidate labels as potential OOD categories during the OOD label mining process, its performance in fine-grained experiments is relatively suboptimal. In contrast, DualCnst demonstrates superior performance, achieving a 1.48% reduction in FPR95 and an 8.56% improvement in AUROC.

Table 7: Additional empirical results with CIFAR-10 and CIFAR-100 as ID datasets. The **bold** indicates the best performance on each dataset.

| ID Dataset | Method | iNaturalist | | SUN | | Places | | Texture | | Average | |
|------------|--------|---------|---------|---------|---------|---------|---------|---------|---------|---------|---------|
| | | FPR95↓ | AUROC↑ | FPR95↓ | AUROC↑ | FPR95↓ | AUROC↑ | FPR95↓ | AUROC↑ | FPR95↓ | AUROC↑ |
| | Energy | 60.70 | 82.12 | 53.14 | 86.00 | 58.29 | 82.86 | 62.52 | 77.89 | 58.66 | 82.22 |
| | MaxLogit | 8.99 | 97.85 | **11.81** | **97.36** | **16.74** | **95.55** | 11.54 | 97.60 | **12.27** | 97.09 |
| CIFAR-10 | MCM | 17.87 | 96.75 | 30.78 | 93.17 | 36.57 | 90.78 | 16.38 | 96.44 | 25.40 | 94.29 |
| | NegLabel | 0.55 | **99.84** | 23.31 | 95.50 | 38.70 | 91.53 | 19.33 | 96.65 | 20.47 | 95.88 |
| | DualCnst | **0.42** | 99.83 | 15.23 | 97.07 | 25.46 | 94.17 | **10.55** | **98.00** | 12.91 | **97.27** |
| | Energy | 82.74 | 74.47 | 67.16 | 81.69 | 68.20 | **80.96** | 81.19 | 66.51 | 74.82 | 75.91 |
| | MaxLogit | 67.77 | 81.41 | 63.26 | 80.72 | 65.73 | 80.81 | 62.94 | 82.00 | 64.93 | 81.24 |
| CIFAR-100 | MCM | 97.95 | 67.50 | 97.69 | 60.71 | 98.40 | 61.34 | 90.23 | 73.58 | 96.07 | 65.78 |
| | NegLabel | 13.95 | 96.47 | 86.61 | 69.04 | 91.50 | 62.08 | 70.60 | 80.26 | 65.66 | 76.96 |
| | DualCnst | **2.88** | **99.23** | 49.35 | 84.25 | 60.68 | 79.06 | 55.35 | 82.65 | 42.07 | 86.30 |
| | Energy | 71.72 | 78.30 | 60.15 | 83.84 | 63.25 | 81.91 | 71.86 | 72.20 | 66.74 | 79.06 |
| | MaxLogit | 38.38 | 89.63 | 37.54 | 89.04 | 41.24 | 88.18 | **37.24** | **89.80** | 38.60 | 89.16 |
| Average | MCM | 57.91 | 82.12 | 64.24 | 76.94 | 67.49 | 76.06 | 53.31 | 85.01 | 60.73 | 80.03 |
| | NegLabel | 7.25 | 98.15 | 54.96 | 82.27 | 65.10 | 76.81 | 44.96 | 88.45 | 43.07 | 86.42 |
| | DualCnst | **1.65** | **99.53** | 32.29 | 95.75 | 37.45 | 90.31 | 45.79 | 86.77 | 27.49 | 91.78 |

Table 8: Zero-shot fine-grained OOD detection results. he **black bold** indicates the best performance. The gray indicates that the comparative methods require training or an additional massive auxiliary dataset.

| Method | ID OOD | CUB-100 CUB-100 | | Stanford-Cars-98 Stanford-Cars-98 | | Food-50 Food-51 | | Oxford-Pet-18 Oxford-Pet-19 | | Average | |
|--------|--------|---------|---------|---------|---------|---------|---------|---------|---------|---------|---------|
| | | FPR95↓ | AUROC↑ | FPR95↓ | AUROC↑ | FPR95↓ | AUROC↑ | FPR95↓ | AUROC↑ | FPR95↓ | AUROC↑ |
| CLIPN | | 73.54 | 74.65 | 53.33 | 82.25 | 43.33 | 88.89 | 53.90 | 86.92 | **56.05** | **83.18** |
| Energy | | 76.13 | 72.11 | 73.78 | 73.82 | 44.95 | 89.97 | 68.51 | 88.34 | 65.84 | 81.06 |
| MaxLogit | | 76.89 | 73.00 | 72.18 | 74.80 | 41.73 | 90.79 | 65.66 | 88.49 | 64.11 | 81.77 |
| MCM | | 83.58 | 67.51 | 83.99 | 68.71 | 43.38 | 91.75 | 63.92 | 84.88 | 68.72 | 78.21 |
| NegLabel | | 82.48 | 68.55 | 79.32 | 70.00 | 37.32 | 92.48 | 66.30 | 88.64 | 66.36 | 79.92 |
| DualCnst | | 77.99 | 72.58 | 78.87 | 70.38 | 36.18 | 92.85 | 66.46 | 88.45 | 64.88 | 81.07 |

# B. Additional Ablation Studies

### B.1. Vision Backbone

This section explores the performance of DualCnst using different CLIP vision encoders.

Table 9 presents the results for ImageNet-1K (ID) with various CLIP vision encoders, including ViT-B/32[1], ViT-L/14[2], RN50[3], RN50x4, RN50x16, and RN101. Across all tested encoders, DualCnst achieves the highest performance. Specifically, compared to ViT-B/16, using ViT-L/14 results in an improvement of 2.33% in FPR95 and 0.37% in AUROC. Furthermore, DualCnst outperforms both zero-shot and fine-tuning methods in OOD detection, achieving the best results in terms of FPR95 and AUROC when utilizing ViT-L/14.

Table 9: Prompt ensembling for text input using different backbones. The ID dataset is ImageNet-1K. The **black bold** indicates the best performance.

| Method | iNaturalist | | SUN | | Places | | Texture | | Average | |
|---|---|---|---|---|---|---|---|---|---|---|
| | FPR95↓ | AUROC↑ | FPR95↓ | AUROC↑ | FPR95↓ | AUROC↑ | FPR95↓ | AUROC↑ | FPR95↓ | AUROC↑ |
| Energy (ViT-B/16) | 79.75 | 83.75 | 79.81 | 83.21 | 70.28 | 83.95 | 88.23 | 71.51 | 79.52 | 80.60 |
| MaxLogit (ViT-B/16) | 67.24 | 87.31 | 66.14 | 86.36 | 61.09 | 85.96 | 80.83 | 76.01 | 68.83 | 83.91 |
| MCM (ViT-B/16) | 40.33 | 92.75 | 35.43 | 92.78 | 44.08 | 89.60 | 54.41 | 87.10 | 43.56 | 90.56 |
| NegLabel (ViT-B/16) | 1.91 | 99.49 | 20.53 | 95.49 | 35.59 | 91.64 | 43.56 | 90.22 | 25.40 | 94.21 |
| DualCnst (ViT-B/16) | **1.29** | **99.65** | **17.60** | **95.89** | **31.91** | **92.13** | **42.15** | **90.51** | **23.24** | **94.55** |
| Energy (ViT-B/32) | 89.22 | 79.15 | 81.01 | 81.62 | 61.22 | 87.20 | 87.64 | 71.36 | 79.77 | 79.83 |
| MaxLogit (ViT-B/32) | 79.45 | 83.75 | 68.89 | 84.85 | 52.30 | 88.60 | 79.88 | 75.29 | 70.13 | 83.12 |
| MCM (ViT-B/32) | 49.81 | 91.37 | 40.31 | 91.80 | 42.94 | 90.08 | 59.33 | 85.32 | 48.10 | 89.64 |
| NegLabel (ViT-B/32) | 3.73 | 99.11 | 22.48 | 95.27 | 34.94 | 91.72 | **50.51** | **88.57** | 27.92 | 93.67 |
| DualCnst (ViT-B/32) | **3.10** | **99.27** | **18.93** | **95.87** | **32.43** | **92.13** | 53.56 | 88.10 | **27.01** | **93.84** |
| Energy (ViT-L/14) | 79.20 | 85.29 | 76.83 | 84.68 | 65.62 | 87.59 | 87.23 | 70.14 | 77.22 | 81.93 |
| MaxLogit (ViT-L/14) | 63.06 | 89.02 | 60.26 | 88.29 | 52.51 | 89.65 | 80.66 | 73.96 | 64.12 | 85.23 |
| MCM (ViT-L/14) | 31.63 | 94.43 | 23.64 | 94.99 | 30.99 | 92.79 | 57.77 | 85.19 | 36.01 | 91.85 |
| NegLabel (ViT-L/14) | 1.77 | 99.53 | 22.33 | 95.63 | 32.22 | 93.01 | 42.92 | 89.71 | 24.81 | 94.47 |
| DualCnst (ViT-L/14) | **1.33** | **99.70** | **19.54** | **96.06** | **26.55** | **93.72** | **42.48** | **89.87** | **22.48** | **94.84** |
| Energy (RN50) | 94.75 | 75.56 | 86.24 | 81.39 | 86.42 | 78.68 | 92.98 | 69.87 | 90.10 | 76.38 |
| MaxLogit (RN50) | 86.45 | 81.21 | 74.56 | 84.31 | 78.15 | 81.10 | 86.45 | 74.61 | 81.40 | 80.31 |
| MCM (RN50) | 45.42 | 91.50 | 43.33 | 91.40 | 55.92 | 86.73 | 55.92 | 86.68 | 50.15 | 89.08 |
| NegLabel (RN50) | 2.88 | 99.24 | 26.51 | 94.54 | 42.60 | 89.72 | 50.80 | 88.40 | 30.70 | 92.97 |
| DualCnst (RN50) | **1.81** | **99.51** | **20.75** | **95.39** | **35.10** | **91.13** | 51.19 | **88.90** | **27.21** | **93.73** |
| Energy (RN50x4) | 85.55 | 81.25 | 80.13 | 84.81 | 68.84 | 85.40 | 92.09 | 69.28 | 81.65 | 80.19 |
| MaxLogit (RN50x4) | 74.51 | 85.14 | 65.51 | 87.61 | 58.86 | 87.26 | 84.47 | 74.81 | 70.84 | 83.70 |
| MCM (RN50x4) | 48.00 | 90.86 | 33.81 | 93.14 | 42.90 | 89.93 | 52.16 | 87.44 | 44.22 | 90.34 |
| NegLabel (RN50x4) | 2.14 | 99.49 | 17.61 | 96.25 | 30.67 | 92.59 | 50.71 | 88.72 | 25.28 | 94.26 |
| DualCnst (RN50x4) | **1.58** | **99.62** | **16.89** | **96.27** | **29.04** | **92.63** | 47.29 | **89.60** | **23.70** | **94.53** |
| Energy (RN50x16) | 73.44 | 86.95 | 65.15 | 88.97 | 73.74 | 83.97 | 84.43 | 76.11 | 74.19 | 84.00 |
| MaxLogit (RN50x16) | 62.10 | 89.05 | 52.35 | 90.45 | 64.74 | 85.69 | 75.66 | 79.37 | 63.71 | 86.14 |
| MCM (RN50x16) | 43.02 | 91.69 | 34.24 | 93.27 | 46.96 | 89.27 | 51.93 | 87.94 | 44.04 | 90.54 |
| NegLabel (RN50x16) | 2.00 | 99.48 | 29.11 | 94.18 | 48.14 | 88.85 | **38.74** | **91.23** | 29.50 | 93.43 |
| DualCnst (RN50x16) | **1.22** | **99.66** | **19.42** | **95.80** | **34.51** | **91.73** | 39.34 | 91.17 | **23.62** | **94.59** |
| Energy (RN101) | 97.82 | 71.11 | 87.81 | 81.10 | 85.43 | 77.92 | 95.96 | 62.32 | 91.75 | 73.11 |
| MaxLogit (RN101) | 92.65 | 77.38 | 74.77 | 84.67 | 75.96 | 81.30 | 90.90 | 68.66 | 83.57 | 78.00 |
| MCM (RN101) | 60.90 | 88.14 | 39.37 | 91.96 | 48.62 | 88.08 | 59.49 | 85.34 | 52.09 | 88.38 |
| NegLabel (RN101) | 2.35 | 99.42 | 21.84 | 95.45 | 41.98 | 90.08 | **53.95** | **87.68** | 30.03 | 93.16 |
| DualCnst (RN101) | 2.56 | 99.36 | **18.93** | **95.88** | **37.52** | **90.89** | 56.03 | 86.88 | **28.76** | **93.26** |

### B.2. Generative Models

To evaluate the effectiveness of DualCnst, different generative models are employed. Table 10 compares the performance of Stable Diffusion v1.5[4] and v2.1[5] using ImageNet-1k as the ID dataset. As Stable Diffusion v2.1 achieves superior generation quality and text-image alignment compared to v1.5, the results under the same random seed 1 show an improvement of 0.6% in FPR95 and 0.14% in AUROC. In terms of generation speed, both models exhibit similar efficiency, requiring

---

[1] https://huggingface.co/openai/clip-vit-base-patch32
[2] https://huggingface.co/openai/clip-vit-large-patch14
[3] https://github.com/openai/CLIP
[4] https://github.com/CompVis/stable-diffusion
[5] https://huggingface.co/stabilityai/stable-diffusion-2-1

approximately 3 seconds per synthetic image.

Table 10: The impact of randomness under different random seeds is examined, with ImageNet-1k as the ID dataset.

| Generative Models | OOD Dataset | | | | | | | | Average | |
| | iNaturalist | | SUN | | Places | | Texture | | | |
| | FPR95↓ | AUROC↑ | FPR95↓ | AUROC↑ | FPR95↓ | AUROC↑ | FPR95↓ | AUROC↑ | FPR95↓ | AUROC↑ |
|---|---|---|---|---|---|---|---|---|---|---|
| stable diffusion v1.5 | 1.27 | 99.65 | 17.30 | 95.94 | 31.61 | 92.15 | 42.91 | 90.32 | 23.27 | 94.51 |
| stable diffusion v2.1 | 1.42 | 99.63 | 17.93 | 95.86 | 32.22 | 92.13 | 39.11 | 90.97 | 22.67 | 94.65 |

## B.3. Encoder Layer

An ablation study was conducted to evaluate the effectiveness of DualCnst using different layers of CLIP's ViT-B/16 encoder. Table 11 presents the results for various layer combinations: (1st, 2nd, 3rd), (4th, 5th, 6th), (7th, 8th, 9th), (9th, 10th, 11th), (3rd, 6th, 9th), and all layers. For each combination, different values of $w$ (0.05, 0.1, 0.15, 0.25) were explored to determine the optimal balance between pixel-level and semantic information.

The results indicate that an equal weight distribution is not necessarily optimal across different layers. For instance, when using the (1st, 2nd, 3rd) layers, setting $w = 0.05$ yields the best performance, as the lower layers primarily capture edge-related features, requiring stronger semantic guidance. In contrast, for the (9th, 10th, 11th) layers, which encode more localized details—such as the fur and ears of a wolf or the stripes of a zebra—assigning a higher weight to visual features leads to improved performance within the DualCnst framework.

Table 11: Using different encoder layers and weights. The ID class labels are the same as ImageNet-1k. The **black bold** indicates the best performance.

| Layer | $w$ | OOD Dataset | | | | | | | | Average | |
| | | iNaturalist | | SUN | | Places | | Texture | | | |
| | | FPR95↓ | AUROC↑ | FPR95↓ | AUROC↑ | FPR95↓ | AUROC↑ | FPR95↓ | AUROC↑ | FPR95↓ | AUROC↑ |
|---|---|---|---|---|---|---|---|---|---|---|---|
| (1st, 2nd, 3rd) | 0.05 | 1.36 | 99.65 | 17.56 | 95.88 | 31.75 | 92.11 | 43.55 | 90.21 | **23.55** | **94.46** |
| | 0.10 | 1.38 | 99.65 | 17.76 | 95.80 | 32.34 | 92.02 | 43.53 | 90.12 | 23.75 | 94.40 |
| | 0.15 | 1.38 | 99.64 | 18.08 | 95.69 | 32.87 | 91.91 | 43.51 | 90.02 | 23.96 | 94.32 |
| | 0.25 | 1.45 | 99.62 | 19.19 | 95.39 | 34.78 | 91.55 | 44.50 | 89.73 | 24.98 | 94.07 |
| (4th, 5th, 6th) | 0.05 | 1.27 | 99.66 | 17.63 | 95.88 | 31.61 | 92.12 | 42.96 | 90.33 | **23.37** | **94.50** |
| | 0.10 | 1.22 | 99.66 | 17.98 | 95.79 | 32.11 | 92.02 | 42.54 | 90.33 | 23.46 | 94.45 |
| | 0.15 | 1.24 | 99.65 | 18.40 | 95.65 | 33.36 | 91.86 | 42.41 | 90.27 | 23.85 | 94.36 |
| | 0.25 | 1.25 | 99.63 | 20.11 | 95.28 | 35.68 | 91.43 | 42.57 | 89.96 | 24.90 | 94.08 |
| (7th, 8th, 9th) | 0.05 | 1.28 | 99.65 | 17.56 | 95.91 | 31.70 | 92.15 | 42.96 | 90.41 | 23.38 | 94.53 |
| | 0.10 | 1.24 | 99.66 | 17.66 | 95.88 | 31.89 | 92.12 | 42.38 | 90.52 | **23.29** | **94.54** |
| | 0.15 | 1.26 | 99.66 | 17.89 | 95.82 | 32.26 | 92.07 | 41.76 | 90.59 | **23.29** | **94.54** |
| | 0.25 | 1.26 | 99.65 | 18.53 | 95.65 | 33.13 | 91.88 | 41.47 | 90.63 | 23.60 | 94.45 |
| (9th, 10th, 11th) | 0.05 | 1.33 | 99.65 | 17.51 | 95.92 | 31.62 | 92.16 | 42.89 | 90.40 | 23.34 | 94.53 |
| | 0.10 | 1.29 | 99.65 | 17.60 | 95.89 | 31.91 | 92.13 | 42.15 | 90.51 | **23.24** | **94.55** |
| | 0.15 | 1.31 | 99.65 | 17.84 | 95.85 | 32.18 | 92.10 | 41.77 | 90.59 | 23.28 | 94.55 |
| | 0.25 | 1.31 | 99.64 | 18.32 | 95.73 | 32.80 | 91.97 | 41.05 | 90.67 | 23.37 | 94.50 |
| (3rd, 6th, 9th) | 0.05 | 1.33 | 99.65 | 17.44 | 95.90 | 31.52 | 92.15 | 43.07 | 90.35 | **23.34** | **94.51** |
| | 0.10 | 1.29 | 99.65 | 17.75 | 95.86 | 32.05 | 92.10 | 42.70 | 90.39 | 23.45 | 94.50 |
| | 0.15 | 1.28 | 99.65 | 17.98 | 95.79 | 32.37 | 92.03 | 42.39 | 90.40 | 23.51 | 94.47 |
| | 0.25 | 1.35 | 99.64 | 18.71 | 95.59 | 33.73 | 91.81 | 42.45 | 90.32 | 24.06 | 94.34 |
| all layer | 0.01 | 1.33 | 99.65 | 17.55 | 95.91 | 31.63 | 92.15 | 43.16 | 90.32 | 23.42 | 94.51 |
| | 0.02 | 1.29 | 99.65 | 17.53 | 95.87 | 31.69 | 92.11 | 42.87 | 90.35 | **23.35** | **94.50** |
| | 0.05 | 1.30 | 99.65 | 18.05 | 95.70 | 32.81 | 91.93 | 42.16 | 90.35 | 23.58 | 94.41 |
| | 0.08 | 1.35 | 99.63 | 19.20 | 95.42 | 34.62 | 91.60 | 42.30 | 90.16 | 24.37 | 94.20 |

## B.4. Fusion Parameter $\alpha$ Of Dual Consistency

This section presents a comprehensive ablation study on the fusion parameter $\alpha$ in the dual consistency method. Experiments are conducted using ImageNet-1k, CIFAR-10, and CIFAR-100 as ID datasets, with iNaturalist, SUN, Places, and Textures serving as OOD datasets. Additionally, experiments are performed by alternately designating ImageNet-10 and ImageNet-20 as ID and OOD datasets.

All experiments utilize the ViT-B/16 visual encoder with selected layers (9th, 10th, 11th) and a fixed weight parameter of

$w = 0.1$. As shown in Table 14, the optimal $\alpha$ value varies across different OOD datasets for ImageNet-1k. Specifically, the best results are obtained with $\alpha = 0.3$ for iNaturalist and Places, $\alpha = 0.1$ for SUN, and $\alpha = 0.2$ for Textures. In the main results, the best-performing $\alpha$ is selected for each OOD dataset. Notably, when $\alpha = 0$, DualCnst reduces to NegLabel.

Table 12 and Table 13 present the results for the CIFAR datasets, where DualCnst consistently outperforms NegLabel. Furthermore, as shown in Table 15, when the ID and OOD datasets exhibit semantic similarities, integrating DualCnst leads to notable performance improvements.

Table 12: An ablation study on the fusion parameter $\alpha$ for cifar10.

| | OOD Dataset | | | | | | | | Average | |
| $\alpha$ | iNaturalist | | SUN | | Places | | Texture | | | |
| | FPR95↓ | AUROC↑ | FPR95↓ | AUROC↑ | FPR95↓ | AUROC↑ | FPR95↓ | AUROC↑ | FPR95↓ | AUROC↑ |
|---|---|---|---|---|---|---|---|---|---|---|
| 0 | 0.55 | 99.84 | 23.31 | 95.5 | 38.7 | 91.53 | 19.33 | 96.65 | 20.47 | 95.88 |
| 0.1 | 0.35 | 99.85 | 17.57 | 96.46 | 30.33 | 93.10 | 12.75 | 97.59 | 15.25 | 96.75 |
| 0.2 | 0.33 | 99.85 | 15.38 | 96.88 | 26.43 | 93.82 | 10.80 | 97.93 | 13.23 | 97.12 |
| 0.3 | 0.42 | 99.83 | 15.23 | 97.07 | 25.46 | 94.17 | 10.55 | 98.00 | 12.91 | 97.27 |
| 0.4 | 0.52 | 99.80 | 15.30 | 97.12 | 25.33 | 94.30 | 10.53 | 97.94 | 12.92 | 97.29 |
| 0.5 | 0.67 | 99.75 | 15.75 | 97.10 | 25.55 | 94.31 | 11.01 | 97.84 | 13.25 | 97.25 |
| 0.6 | 0.89 | 99.67 | 16.18 | 97.04 | 26.17 | 94.24 | 11.49 | 97.71 | 13.68 | 97.17 |
| 0.7 | 1.39 | 99.55 | 16.74 | 96.95 | 26.89 | 94.14 | 11.95 | 97.59 | 14.24 | 97.06 |
| 0.8 | 1.97 | 99.38 | 17.14 | 96.86 | 27.49 | 94.03 | 12.27 | 97.47 | 14.72 | 96.94 |
| 0.9 | 3.13 | 99.16 | 17.71 | 96.76 | 27.87 | 93.91 | 12.68 | 97.37 | 15.35 | 96.80 |
| 1 | 4.74 | 98.88 | 18.27 | 96.66 | 28.26 | 93.79 | 13.14 | 97.26 | 16.10 | 96.65 |

Table 13: An ablation study on the fusion parameter $\alpha$ for cifar100.

| | OOD Dataset | | | | | | | | Average | |
| $\alpha$ | iNaturalist | | SUN | | Places | | Texture | | | |
| | FPR95↓ | AUROC↑ | FPR95↓ | AUROC↑ | FPR95↓ | AUROC↑ | FPR95↓ | AUROC↑ | FPR95↓ | AUROC↑ |
|---|---|---|---|---|---|---|---|---|---|---|
| 0 | 13.95 | 96.47 | 86.61 | 69.04 | 91.5 | 62.08 | 70.6 | 80.26 | 65.66 | 76.96 |
| 0.1 | 11.63 | 97.18 | 83.36 | 70.68 | 89.27 | 64.39 | 61.79 | 83.27 | 61.51 | 78.88 |
| 0.2 | 9.55 | 97.72 | 80.42 | 72.56 | 87.11 | 66.52 | 57.20 | 84.67 | 58.57 | 80.37 |
| 0.3 | 7.51 | 98.11 | 75.99 | 74.40 | 83.68 | 68.46 | 54.77 | 85.01 | 55.49 | 81.50 |
| 0.4 | 6.40 | 98.42 | 72.84 | 76.14 | 79.90 | 70.26 | 55.48 | 84.72 | 53.65 | 82.39 |
| 0.5 | 5.29 | 98.66 | 67.84 | 77.83 | 75.92 | 72.01 | 55.62 | 84.22 | 51.17 | 83.18 |
| 0.6 | 4.40 | 98.85 | 63.12 | 79.45 | 72.40 | 73.70 | 55.12 | 83.74 | 48.76 | 83.94 |
| 0.7 | 3.80 | 99.00 | 58.51 | 80.94 | 68.65 | 75.30 | 54.96 | 83.38 | 46.48 | 84.66 |
| 0.8 | 3.33 | 99.11 | 54.69 | 82.25 | 65.38 | 76.75 | 54.73 | 83.10 | 44.53 | 85.30 |
| 0.9 | 2.97 | 99.18 | 51.49 | 83.35 | 62.70 | 78.00 | 55.12 | 82.86 | 43.07 | 85.85 |
| 1 | 2.88 | 99.23 | 49.35 | 84.25 | 60.68 | 79.06 | 55.35 | 82.65 | 42.07 | 86.30 |

Table 14: An ablation study on the fusion parameter $\alpha$ for ImageNet-1k.

| | OOD Dataset | | | | | | | | Average | |
| $\alpha$ | iNaturalist | | SUN | | Places | | Texture | | | |
| | FPR95↓ | AUROC↑ | FPR95↓ | AUROC↑ | FPR95↓ | AUROC↑ | FPR95↓ | AUROC↑ | FPR95↓ | AUROC↑ |
|---|---|---|---|---|---|---|---|---|---|---|
| 0 | 1.91 | 99.49 | 20.53 | 95.49 | 35.59 | 91.64 | 43.56 | 90.22 | 25.40 | 94.21 |
| 0.1 | 1.29 | 99.65 | 17.60 | 95.89 | 31.91 | 92.13 | 42.15 | 90.51 | 23.24 | 94.55 |
| 0.2 | 1.09 | 99.68 | 18.15 | 95.77 | 31.79 | 91.93 | 42.00 | 90.32 | 23.26 | 94.42 |
| 0.3 | 0.99 | 99.69 | 18.49 | 95.67 | 31.63 | 91.73 | 43.40 | 89.85 | 23.63 | 94.23 |
| 0.4 | 0.99 | 99.67 | 18.97 | 95.33 | 32.29 | 91.39 | 46.13 | 89.07 | 24.60 | 93.87 |
| 0.5 | 1.00 | 99.64 | 19.93 | 95.13 | 33.40 | 91.08 | 48.78 | 88.37 | 25.78 | 93.56 |
| 0.6 | 1.12 | 99.59 | 20.83 | 94.90 | 34.38 | 90.77 | 51.65 | 87.64 | 26.99 | 93.22 |
| 0.7 | 1.43 | 99.52 | 21.89 | 94.66 | 35.43 | 90.46 | 53.60 | 86.87 | 28.09 | 92.88 |
| 0.8 | 1.73 | 99.42 | 22.79 | 94.41 | 36.04 | 90.17 | 54.86 | 86.08 | 28.85 | 92.52 |
| 0.9 | 2.08 | 99.29 | 23.67 | 94.15 | 36.59 | 89.88 | 56.56 | 85.27 | 29.73 | 92.15 |
| 1 | 2.86 | 99.12 | 24.72 | 93.89 | 37.60 | 89.59 | 58.30 | 84.42 | 30.87 | 91.75 |

## B.5. The Randomness Of Stable Diffusion

An ablation study is conducted to assess the impact of Stable Diffusion's randomness on the effectiveness of DualCnst in generating synthetic images. Specifically, synthetic images are generated using three different random seeds, and the results are evaluated on ImageNet-1k. The experiments employ the ViT-B/16 visual encoder with selected layers (9th, 10th, 11th), along with fixed parameters $w = 0.1$ and $\alpha = 0.1$.

Table 15: An ablation study on the parameter $\alpha$, alternating ImageNet10 and ImageNet20 as ID and OOD datasets.

| $\alpha$ | ID / OOD | ImageNet-10 / ImageNet-20 | | ImageNet-20 / ImageNet-10 | | **Average** | |
|---|---|---|---|---|---|---|---|
| | | **FPR95↓** | **AUROC↑** | **FPR95↓** | **AUROC↑** | **FPR95↓** | **AUROC↑** |
| 0.0 | | 5.10 | 98.86 | 17.60 | 97.04 | 11.35 | 97.95 |
| 0.1 | | 2.20 | 98.77 | 13.60 | 97.61 | 8.15 | 98.31 |
| 0.2 | | 3.10 | 98.98 | 12.20 | 97.44 | 7.65 | 98.21 |
| 0.3 | | 4.00 | 98.89 | 18.60 | 97.06 | 11.30 | 97.98 |
| 0.4 | | 4.90 | 98.72 | 18.40 | 97.09 | 11.65 | 97.91 |
| 0.5 | | 5.30 | 98.54 | 18.00 | 97.12 | 11.65 | 97.83 |
| 0.6 | | 6.40 | 98.33 | 17.80 | 97.16 | 12.10 | 97.75 |
| 0.7 | | 8.60 | 98.07 | 16.80 | 97.21 | 12.70 | 97.64 |
| 0.8 | | 10.30 | 97.78 | 15.40 | 97.41 | 12.85 | 97.60 |
| 0.9 | | 12.60 | 97.47 | 13.60 | 97.65 | 13.10 | 97.56 |
| 1 | | 13.10 | 97.14 | 48.00 | 97.65 | 30.55 | 97.40 |

As shown in Table 16, the performance remains consistent across different random seeds, indicating that the inherent randomness of Stable Diffusion does not significantly impact the effectiveness of DualCnst.

Table 16: The impact of randomness under different random seeds is examined, with ImageNet-1k as the ID dataset.

| Random | **OOD Dataset** iNaturalist | | SUN | | Places | | Texture | | **Average** | |
|---|---|---|---|---|---|---|---|---|---|---|
| | **FPR95↓** | **AUROC↑** | **FPR95↓** | **AUROC↑** | **FPR95↓** | **AUROC↑** | **FPR95↓** | **AUROC↑** | **FPR95↓** | **AUROC↑** |
| random 1 | 1.27 | 99.65 | 17.30 | 95.94 | 31.61 | 92.15 | 42.91 | 90.32 | 23.27 | 94.51 |
| random 2 | 1.29 | 99.65 | 17.60 | 95.89 | 31.91 | 92.13 | 42.15 | 90.51 | 23.24 | 94.55 |
| random 3 | 1.26 | 99.66 | 18.16 | 95.64 | 32.26 | 91.90 | 41.90 | 90.33 | 23.39 | 94.38 |

## B.6. Score Function

We present the specific form of the score function designed in the ablation study. They are $S_{\text{MAX}}$, $S_{\text{Energy}}$ and $S_{\text{MaxLogit}}$. Firstly, we review the definition of the fused visual-text cosine similarity $\tilde{s}$ as:

$$\tilde{s}_i(\mathbf{x}) = \alpha \cdot s_{i,\text{img}}(\mathbf{x}) + (1-\alpha) \cdot s_{i,\text{text}}(\mathbf{x}) \tag{7}$$

where

$$s_{i,\text{img}}(\mathbf{x}) = \sum_{l=1}^{L} w_l \cdot s_{i,\text{img}}^{(l)}(\mathbf{x})$$

with

$$s_{i,\text{img}}^{(l)}(\mathbf{x}) = \frac{\mathcal{I}^{(l)}(\mathbf{x}) \cdot \mathcal{I}^{(l)}(\bar{\mathbf{x}}_i)}{\|\mathcal{I}^{(l)}(\mathbf{x})\| \cdot \|\mathcal{I}^{(l)}(\bar{\mathbf{x}}_i)\|}, \quad \bar{\mathbf{x}}_i \in \bar{\mathcal{X}} \tag{8}$$

and

$$s_{i,\text{text}}(\mathbf{x}) = \frac{\mathcal{I}(\mathbf{x}) \cdot \mathcal{T}(\mathbf{t}_i)}{\|\mathcal{I}(\mathbf{x})\| \cdot \|\mathcal{T}(\mathbf{t}_i)\|} \tag{9}$$

The specific form of $S_{\text{MAX}}$ is as follows:

$$S_{\text{MAX}}(\mathbf{x}; \mathcal{Y}^{\text{id}} \cup \mathcal{Y}^{\text{ood}}, \bar{\mathcal{X}}, \mathcal{T}, \mathcal{I}) = \begin{cases} \frac{1}{K}, & \max\limits_{i \in [1,K]} \tilde{s}_i < \max\limits_{j \in [K+1, K+M]} \tilde{s}_j, \\ \max\limits_{i \in [1,K]} \frac{e^{\tilde{s}_i(\mathbf{x})}}{\sum_{j=1}^{K} e^{\tilde{s}_j(\mathbf{x})}}, & \max\limits_{i \in [1,K]} \tilde{s}_i \geq \max\limits_{j \in [K+1, K+M]} \tilde{s}_j. \end{cases} \tag{10}$$

$S_{\text{MAX}}$ indicates that if the $\tilde{s}_j$ $(j \in [K+1, K+M])$ of an input sample is larger than the $\tilde{s}_i$ $(i \in [1, K])$, this sample is recognized to be an OOD sample. This implies that the maximum similarity observed between the input sample and any OOD visual-text similarity exceeds the similarity between the input sample and any ID visual-text similarity. Otherwise, the input sample is evaluated based on the maximum softmax probability.

Similarly, $S_{\text{Energy}}$ and $S_{\text{MaxLogit}}$ are modifications of the Energy and MaxLogit metrics, respectively, incorporating visual-text similarity into their secondary components.

$$S_{\text{Energy}}(\mathbf{x}; \mathcal{Y}^{\text{id}} \cup \mathcal{Y}^{\text{ood}}, \bar{\mathcal{X}}, \mathcal{T}, \mathcal{I}) = -T \left( \log \sum_{i=1}^{K} e^{\tilde{f}_i(\mathbf{x})/T} - \log \sum_{j=K+1}^{K+M} e^{\tilde{f}_j(\mathbf{x})/T} \right), \tag{11}$$

$$S_{\text{MaxLogit}}(\mathbf{x}; \mathcal{Y}^{\text{id}} \cup \mathcal{Y}^{\text{ood}}, \bar{\mathcal{X}}, \mathcal{T}, \mathcal{I}) = \max_{i \in [1, K]} \tilde{s}_i(\mathbf{x}) - \max_{j \in [K+1, K+M]} \tilde{s}_j(\mathbf{x}). \tag{12}$$

Table 17 presents the detailed experimental results on ImageNet-1k (ID).

Table 17: Additional ablation studies on score functions. The **bold** indicates the best performance on each dataset.

| Score Funtion | iNaturalist | | SUN | | Places | | Texture | | Average | |
|---|---|---|---|---|---|---|---|---|---|---|
| | FPR95↓ | AUROC↑ | FPR95↓ | AUROC↑ | FPR95↓ | AUROC↑ | FPR95↓ | AUROC↑ | FPR95↓ | AUROC↑ |
| $S_{\text{MAX}}$ | 100.00 | 83.00 | 100.00 | 82.16 | 100.00 | 80.62 | 100.00 | 80.28 | 100.00 | 81.51 |
| $S_{\text{Energy}}$ | 1.98 | 99.42 | 20.26 | 95.52 | 35.54 | 91.49 | 45.69 | 89.96 | 25.87 | 94.10 |
| $S_{\text{MaxLogit}}$ | 6.26 | 98.58 | 29.54 | 93.79 | 43.18 | 89.55 | 50.78 | 87.95 | 32.44 | 92.47 |
| $S_{\text{DualCnst}}$ | **0.99** | **99.69** | **17.60** | **95.89** | **31.63** | **91.73** | **42.00** | **90.32** | **23.05** | **94.41** |

## C. Experimental Configuration and Details

### C.1. Details of Mining Potential OOD Labels

Before generating synthetic images, it is crucial to identify effective OOD labels by leveraging ID labels as a reference. Specifically, we define the set of ID labels as $\mathcal{Y}^{\text{id}} = \{y_1, y_2, \ldots, y_K\}$ and collect a pool of nouns and adjectives from open-world resources (e.g., WordNet (Fellbaum, 1998), ConceptNet (Speer et al., 2017), and Wikipedia Categories (wik, 2023)) as candidate OOD labels, denoted by $\mathcal{Y}^{\text{c}} = \{\tilde{y}_1, \tilde{y}_2, \ldots, \tilde{y}_C\}$, where $C$ represents the total number of candidates.

To assess the semantic relationship between candidate OOD labels and ID labels, we utilize CLIP's text encoder to extract text embeddings for both sets. The embedding of a candidate OOD label is given by $\tilde{\mathbf{e}}_c = \mathcal{T}(\text{prompt}(\tilde{y}_c))$, while the embedding of an ID label is represented as $\mathbf{e}_k = \mathcal{T}(\text{prompt}(y_k))$. By default, we employ the prompt format `"A photo of a <label>"` to generate these embeddings.

Following the methodology outlined in NegMining (Jiang et al., 2024), we quantify the semantic distance between each candidate OOD label and the ID labels using negative cosine similarity. Specifically, for a given candidate OOD label, we compute its negative cosine similarity with all ID label embeddings, resulting in $K$ similarity scores. The overall semantic distance of an OOD label to the ID label set is then determined as the $\eta$-percentile (default $\eta = 0.05$) of these scores:

$$d_c = \text{percentile}_\eta \left( \left\{ -\cos(\tilde{\mathbf{e}}_c, \mathbf{e}_k) \right\}_{k=1}^{K} \right). \tag{13}$$

After computing distances for all candidate OOD labels, we select the top $M = 10,000$ labels with the greatest distances. The selected OOD label set is defined as:

$$\mathcal{Y}^{\text{ood}} = \text{TopK}\left( \{d_c\}_{c=1}^{C}, \mathcal{Y}^{\text{c}}, M \right). \tag{14}$$

During the generation phase, DualCnst utilizes $\mathcal{Y}^{\text{ood}} \cup \mathcal{Y}^{\text{id}}$ as the label space for synthetic image generation. To ensure semantic consistency, it employs stable diffusion to generate images that align with these labels, thereby providing meaningful visual representations to enhance the inference process.

### C.2. Evaluation Metrics

In this study, we adopt the most widely used evaluation metrics in the OOD detection domain, including FPR95 and AUROC (Yang et al., 2022). To further assess the effectiveness of the proposed dual consistency method under additional evaluation criteria, we also report AUPR results for CLIP-B/16 in Table 18. The results demonstrate that our dual consistency method achieves superior performance across all evaluation metrics.

Table 18: Performance in terms of AUPR. ID dataset: The experiments are zero-shot OOD detection results with ImageNet-1K as the ID dataset. The **black bold** indicates the best performance. The gray indicates that the comparative methods require training or an additional massive auxiliary dataset.

| Method | OOD Dataset | | | | Average |
| | iNaturalist | SUN | Places | Texture | |
|---|---|---|---|---|---|
| CLIPN | 99.15 | 98.59 | **98.22** | 98.38 | 98.59 |
| Energy | 96.84 | 96.50 | 96.16 | 94.66 | 96.04 |
| MaxLogit | 97.74 | 97.12 | 96.65 | 95.61 | 96.78 |
| MCM | 98.86 | 98.28 | 97.49 | 98.04 | 98.17 |
| NegLabel | 99.80 | 98.79 | 97.76 | 98.08 | 98.61 |
| DualCnst | **99.92** | **99.02** | 98.01 | **98.72** | **98.92** |

## C.3. Experimental Configuration

This paper introduces a dual consistency (DualCnst) method, implemented using Python 3.8 and PyTorch 1.13 library (Paszke et al., 2019), with all experiments conducted on a single NVIDIA RTX A6000 GPU. Prior to experimentation, the proposed method generates synthetic images, with each image requiring approximately 3 seconds for generation. To mitigate redundant computational overhead across multiple runs, we precompute and store the visual features of the generated images.

In Table 19, we report the computational cost of DualCnst in generating synthetic images based on NegLabel (Jiang et al., 2024) and provide a comparative analysis of inference times. In ImageNet-1k experiments, DualCnst requires 10 hours and 22 minutes to generate one synthetic image per label, while inference takes 17 minutes. In comparison, NegLabel requires 14 minutes and 35 seconds for inference on ImageNet-1k. These results demonstrate that DualCnst does not introduce excessive computational overhead during inference.

For the selection of negative label parameters, we adopt the optimal configuration recommended in NegLabel. All experiments in this study are conducted within the CLIP framework. Unless otherwise specified, we utilize CLIP-B/16 for zero-shot OOD detection. The default hyperparameter settings are as follows: We set $w = 0.1$ and extract intermediate-layer features from the 9th, 10th, and 11th layers of the visual encoder, which are then fused with the final semantic features. The sum-softmax score is employed, with the fusion parameter set to $\alpha = 0.1$ and the temperature parameter to $\tau = 0.01$.

Table 19: Computational cost of DualCnst and NegLabel on ImageNet-1k.

| Method | Image Generation Time | Inference Time |
|---|---|---|
| DualCnst | 10h 22m | 17m |
| NegLabel (Jiang et al., 2024) | - | 14m 35s |

