# OpenReview forum: "Text-Image Dual Consistency-Guided OOD Detection with Pretrained Vision-Language Models"
_ICML.cc/2025/Conference — Submitted to ICML 2025_

### Official Review · Reviewer_aTAh · 2025-02-18

**Overall Recommendation:** 2

**Summary:**

This paper introduces DualCnst, a novel text-image dual consistency framework for zero-shot Out-of-Distribution (OOD) detection using pretrained Vision-Language Models (VLMs) like CLIP. The core idea is to leverage both semantic similarity (text-based) and visual similarity (image-based) by generating synthetic ID/OOD images via Stable Diffusion. The proposed approach outperforms state-of-the-art (SOTA) methods such as NegLabel and MCM on multiple OOD detection benchmarks, demonstrating superior generalization across diverse datasets.

**Claims And Evidence:**

Yes

**Essential References Not Discussed:**

No

**Experimental Designs Or Analyses:**

Yes

**Methods And Evaluation Criteria:**

Yes

**Other Comments Or Suggestions:**

Comments:
1. The paper assumes that Stable Diffusion can reliably supplement visual information for ID/OOD detection. However, this assumption may not always hold:
(1) Stable Diffusion may generate images with noise, style variations, or artifacts, potentially misleading the OOD detection model.
(2) The paper does not analyze how synthetic images compare with real ID data. If the generated images significantly deviate from the true ID distribution, they may negatively impact OOD detection performance.
(3) The paper does not assess whether the generated images faithfully represent ID/OOD distributions or if they introduce biases that could mislead the detection process.
2. The proposed DualCnst score function is a simple weighted combination of textual and visual similarities, but there is no theoretical analysis to justify the chosen weighting strategy.
3. The paper does not provide a mathematical proof of convergence or generalization for the score function, relying only on empirical validation. A more rigorous theoretical foundation would strengthen the contribution.
4. The paper provides a discussion on computational cost in Table 19, reporting that DualCnst requires 10 hours and 22 minutes to generate synthetic images, while inference takes 17 minutes, compared to NegLabel’s 14 minutes and 35 seconds for inference on ImageNet-1k. While the inference overhead is reasonable, the preprocessing time for image generation is significantly high, which may limit the practicality of the approach in real-world applications. The authors should explore ways to reduce computational overhead.

**Other Strengths And Weaknesses:**

Strengths:
1. The paper proposes a new multimodal OOD detection approach, integrating textual and visual consistency.
Unlike existing methods (e.g., NegLabel), which focus only on text-based features, DualCnst explores the underutilized visual similarity.
2. The approach achieves state-of-the-art (SOTA) performance on standard benchmarks, with notable improvements in far OOD (+2.35%), near OOD (+3.9%), and robust OOD detection (+9.9%).

Weaknesses:

See comments.

**Questions For Authors:**

Questions:
1. Stable Diffusion performs well on natural image datasets, but is it still effective for OOD detection in domains such as medical imaging or remote sensing? Have the authors considered evaluating its generalization across different types of datasets?

**Relation To Broader Scientific Literature:**

No

**Theoretical Claims:**

Yes

---

> ### Author Rebuttal · Authors · 2025-04-01
>
> Response to Reviewer aTAh
> We thank the reviewer aTAh for the valuable feedback. We addressed all the comments. Please find the point-to-point responses below. Any further comments and discussions are welcomed\!
> **W1:** The paper assumes that Stable Diffusion can reliably supplement visual information for ID/OOD detection. However, this assumption may not always hold: (1)**Generation Quality**(2)**Distribution Alignment**(3) **Representation Fidelity**
> **Reply:**
> **R1:** We appreciate the suggestion. While synthetic images may contain noise, style variations, or artifacts that could potentially mislead OOD detection models, our score function incorporates multi-level features from pixel to semantic levels. Thus, even when pixel-level features are affected by such deviations, language-level features remain effective. To validate this, we generated **oil painting-style** images to simulate style shifts. As shown in Table A, our method maintains robust performance despite this variation.  Due to space constraints, the complete table is available at:
> https://anonymous.4open.science/r/fjutlfy-31D8 Table K.
>
> Table A: Experimental comparison under generated image style shifts. ID dataset: ImageNet-1k.
>
> | Image Style (Method) | Average |
> | :---: | ----- |
> |  | FPR95 |
> | Natural Images (DualCnst) | **23.24/94.55** |
> | Oil Painting Images (DualCnst) ） | 23.83/94.37 |
> | NegLabel | 25.40/94.21 |
>
> **R2 and R3:** We confirm that the synthetic ID/OOD data maintain consistent language-level features with real data, ensuring the robustness of our method (as validated above) . Notably, as demonstrated in Experimrnts (Table A) in R1, we deliberately generated oil-style images that deviate from the ID distribution, yet they still contribute to improved detection performance. This further demonstrates the robustness of language-level features.
>
> **W2:** The proposed DualCnst score function is a simple weighted combination of textual and visual similarities, but there is no theoretical analysis to justify the chosen weighting strategy.
> **Reply:**
> We gratefully acknowledge the reviewer's suggestion. Our supplemented theoretical analysis demonstrates that under a weighted combination strategy of textual and visual similarities, the false positive rate ($\\text{FPR}\_\\lambda$) exhibits a monotonic decrease with increasing multimodal (visual) labels under certain conditions. This result not only validates that incorporating auxiliary visual features enhances OOD detection performance, but also confirms the appropriateness of the weighted combination strategy. Due to space limitations, the detailed theoretical analysis can be found in the response to Reviewer YYQ7-W1.
> **W3:** The paper does not provide a mathematical proof of convergence or generalization for the score function, relying only on empirical validation. A more rigorous theoretical foundation would strengthen the contribution.
> **Reply:**
> As noted in our W2 reply, we have discussed the proposed score function's ability to improve the separability between ID and OOD samples in terms of the false positive rate ($\\text{FPR}\_\\lambda$). Specifically, the proposed score function integrates multi-level synthetic image features into the existing text-based labels. Theoretically, we consider a more general scenario—how expanding multimodal labels improves OOD detection. We prove that, under certain conditions, $\\text{FPR}\_\\lambda$ decreases as the number of multimodal labels increases, demonstrating that incorporating additional auxiliary modalities into labels improves OOD detection performance.
> **W4:** DualCnst's synthetic image generation incurs significantly higher costs than baselines, potentially limiting real-world applicability. Urge optimization of computational overhead.
> **Reply:**
> Thank you for your comment. Regarding the efficiency of Stable Diffusion generation, we address the issue through two key optimizations.Due to space limitations, Reviewer ViPC raised the same question. Please refer to our detailed response under W5 in the **Reviewer ViPC** section.
> **Q:** Stable Diffusion performs well on natural image datasets, but is it still effective for OOD detection in domains such as medical imaging or remote sensing? Have the authors considered evaluating its generalization across different types of datasets?
> **Reply:**
> We have supplemented experiments on remote sensing data using UCM\[1\] as the ID dataset and AID\[2\] as the OOD dataset. As shown in Table D, our method achieves superior performance compared to the baseline NegLabel.
> \[1\]Y. Yang and S. Newsam, "Bag-of-visual-words and spatial extensions for land-use classification "
> \[2\]Zhong et al. "AID: A benchmark dataset for performance evaluation of aerial scene classification "
> Table D: OOD Detection Experiments on Remote Sensing Data.
>
> | Method | AID |
> | :---: | :---: |
> |  | FPR95 |
> | NegLabel | 97.98/62.46 |
> | DualCnst(ours) | **97.19/66.11** |

---

### Official Review · Reviewer_Ci5u · 2025-03-10

**Overall Recommendation:** 3

**Summary:**

This paper propose DualCnst for CLIP-based zero-shot OOD detection. It enhances zero-shot OOD detection by combining text-image dual consistency, leveraging both semantic similarity to textual labels and visual similarity to synthesized images. This unified framework achieves state-of-the-art performance across benchmarks and generalizes well across VLM architectures.

**Claims And Evidence:**

The main claim of this paper is that
1. adding visual feature to CLIP-based zero-shot OOD detection can improve the performance.
2. The proposed method out-perform previous sota.

By properly introduce the visual feature from CLIP-based models with stable diffusion, this method work well and out-perform previous sota through extensive experiments.

**Essential References Not Discussed:**

6. While L421 mentioned VAE-based OOD detection methods, they are pretty old. It would be better to discuss relationship with some new methods based on generative models, such as [a] with GAN and [b-c] with diffusion model. Especially, [b-c] also use stable diffusion and perform training-free OOD detection, should also be discussed in the last paragraph of Sec. 5.

[a] Out-of-Distribution Detection with Semantic Mismatch under Masking, Yang et al. ECCV 2022.

[b] DiffGuard: Semantic Mismatch-Guided Out-of-Distribution Detection using Pre-trained Diffusion Models, Gao et al. ICCV 2023.

[c] Outof-distribution detection with diffusion-based neighborhood, Liu et al. 2023.

**Experimental Designs Or Analyses:**

3. Considering negative images, is the upper bound given by using real OOD samples? The authors are encouraged to include this as one of the ablation studies.
4. The training of CLIP and stable diffusion both involves large amout of data, which may cover the OOD datasets. How to ensure fairness in evaluation? Especially consider that the authors have differentiated some methods using massive auxilary dataset in Table 1 & 3.

**Methods And Evaluation Criteria:**

Most descriptions are clear to me. Just a few questions leaving:
1. Why ID images are not accessible (fig. 1)? Please justify. I think most testing benchmarks provide training samples which can be used as "actual ID Image". This scenario should be consider as a baseline.
2. Is it necessary to have negative labels/images? If it is necessary, it would be hard to judge the contribution of this paper over NegLabel. The authors are encouraged to include an ablation study to emphasize image feature without using negative labels/images.

**Other Comments Or Suggestions:**

10. The authors have talked about different CLIP models but the architectures for diffusion models are similar. The authors are encouraged to consider other models such as DiT-based ones.

**Other Strengths And Weaknesses:**

**Strengths**:
7. This article conducts comprehensive experimental comparisons on multiple benchmarks, demonstrating the effectiveness of the proposed method.
8. The paper is well-written and easy to follow.

**Weaknesses**:
9. I noticed that the image generation time is pretty long in Table 19, which should be emphasized in the main paper as one of the limitations.

**Questions For Authors:**

Please see above.

**Relation To Broader Scientific Literature:**

5. OOD detection is highly valuable for understanding and modeling the boundaries of models. The method proposed in this paper effectively leverages the text and image modalities provided by the CLIP model, combined with generative models, offering significant insights for the future development of OOD detection in the era of large models.

**Theoretical Claims:**

There is no theoretical claims.

---

> ### Author Rebuttal · Authors · 2025-04-01
>
> Response to Reviewer Ci5u
> We thank the reviewer Ci5u for the valuable feedback. We addressed all the comments. Please find the point-to-point responses below. Any further comments and discussions are welcomed\!
> **W1:** Why ID images are not accessible (fig. 1\) ? Please justify. I think most testing benchmarks provide training samples which can be used as "actual ID Image". This scenario should be consider as a baseline.
> **Reply:**
> Thank you for your question. However, in the **zero-shot** OOD detection setting considered in this paper, training samples are not ID images. Specifically, the definitions of ID and OOD labels are as follows. As stated in the **Introduction section of ZOC**.
> **W2:** If negative labels are essential, the method’s novelty over NegLabel is unclear. An ablation study (removing negatives) is needed to isolate the impact of synthetic image features.
> **Reply:**
> Our method is not limited to applications with NegLabel. As shown in Table A, the results demonstrate that incorporating synthetic image labels can effectively enhance OOD detection performance even without negative labels/images. Due to space constraints, the complete table is available at:
> https://anonymous.4open.science/r/fjutlfy-31D8 Table H.
> Table A: Performance comparison of integrating DualCnst with MCM for OOD detection on ImageNet-1k (ID dataset) .
>
> | SD model | Average |
> | :---: | ----- |
> |  | FPR95 |
> | MCM | 65.20 |
> | MCM+DualCnst | **63.73** |
>
> **W3:** Considering negative images, is the upper bound given by using real OOD samples? The authors are encouraged to include this as one of the ablation studies.
> **Reply:**
> Yes, we have conducted ablation studies using real OOD samples (Table B) . The results demonstrate that while the proposed method achieves better performance with real samples compared to synthetic ones, the improvement margin is not substantial, demonstrating synthetic samples' effectiveness. Due to space constraints, the complete table is available at:
> https://anonymous.4open.science/r/fjutlfy-31D8 Table I.
> Table B: Experimental comparison between real OOD samples and synthetic OOD samples, with ImageNet-1k as the ID dataset.
>
> | Source of OOD images | Average |
> | :---: | ----- |
> |  | FPR95 |
> | Synthetic OOD samples | 23.24 |
> | Real OOD samples | **12.36** |
>
> **W4:** How to ensure evaluation fairness for models (e. g. , CLIP/Stable Diffusion) trained on large datasets that may include OOD data, particularly when comparing methods with/without auxiliary datasets?
> **Reply:**
> Thanks for your question. We would like to clarify our definition of ID classes. Following zero-shot OOD detection \[1, 2, 3\], in our setting, the ID classes are defined based on the classification task of interest rather than the classes used in pre-training. Additionally, we adopt widely recognized zero-shot OOD detection benchmarks, where the label spaces of ID and OOD datasets do not overlap.
> \[1\] Ming et al. Delving into Out-of-Distribution Detection with Vision-Language Representations.
> \[2\] Jiang et al. Negative Label Guided OOD Detection with Pretrained Vision-Language Models.
> \[3\] Esmaeilpour et al. Zero-Shot Out-of-Distribution Detection Based on the Pre-trained Model CLIP.
> **W5:** Suggest expanding discussion in Sec. 5 to include newer GAN/diffusion-based OOD methods (e. g. , \[a\] with GAN, \[b-c\] with training-free Stable Diffusion) rather than older VAE-based approaches.
> **Reply:**
> Thank you for your suggestions. These three articles will be incorporated into the Related Work section.
> **W6:** I noticed that the image generation time is pretty long in Table 19, which should be emphasized in the main paper as one of the limitations.
> **Reply:**
> We acknowledge this as a limitation of our current framework and will include it in the limitations. However, we note that the computational efficiency can be significantly optimized without compromising detection performance.
> Due to space limitations, Reviewer ViPC raised the same question. Please refer to our detailed response under W5 in the **Reviewer ViPC** section.
> **W7:** The authors have talked about different CLIP models but the architectures for diffusion models are similar. The authors are encouraged to consider other models such as DiT-based ones.
> **Reply:**
> We appreciate the suggestion. we incorporate synthetic images generated by Hunyuan-DiT \[1\] in our experiments, compared to SD1.5, Hunyuan-DiT achieved better results on the FPR95 metric.Due to space constraints, the complete table is available at:
> https://anonymous.4open.science/r/fjutlfy-31D8 Table J.
> \[1\] Hunyuan-DiT: A Powerful Multi-Resolution Diffusion Transformer with Fine-Grained Chinese Understanding.
> Table E: Performance comparison between SD model and DiT models in the DualCnst method.
>
> | SD model | Average |
> | :---: | :---: |
> |  | FPR95 |
> | SD1. 5 | 23.24 |
> | Hunyuan-DiT | **23.19** |

---

### Official Review · Reviewer_yyQ7 · 2025-03-12

**Overall Recommendation:** 3

**Summary:**

This paper presents a simple and effective method to enhance the performance of OOD detection. In addition to utilizing the similarity between test images and text features, it also introduces images through a diffusion model, thereby leveraging the similarity between test images and generated images to further improve OOD detection performance.

**Claims And Evidence:**

Yes

**Essential References Not Discussed:**

The main contribution of this paper is to enhance the zero-shot OOD detection capability of the CLIP model. Recently, there have been many related papers in this direction, such as [1,2]. It is suggested that the authors add discussion and comparison.
[1] LAPt: Label-driven Automated Prompt Tuning for OOD Detection with Vision-Language Models, ECCV2024
[2] CLIPScope: Enhancing Zero-Shot OOD Detection with Bayesian Scoring

**Experimental Designs Or Analyses:**

Yes

**Methods And Evaluation Criteria:**

Yes

**Other Comments Or Suggestions:**

NA

**Other Strengths And Weaknesses:**

The biggest advantage of this paper is that the method is simple and effective. However, this might also be its biggest disadvantage: because the contribution of the technique is relatively small. Therefore, it is recommended that the authors add some profound explanations, such as theoretical support or vivid visual demonstrations, to enrich the content.

**Questions For Authors:**

See questions above.

**Relation To Broader Scientific Literature:**

NA

**Theoretical Claims:**

NA

---

> ### Author Rebuttal · Authors · 2025-04-01
>
> Response to Reviewer yyQ7
> We thank the reviewer yyQ7 for the valuable feedback. We addressed all the comments. Please find the point-to-point responses below. Any further comments and discussions are welcomed\!
> **W1:** While the method’s simplicity is a strength, it risks underselling technical novelty. Expand theoretical guarantees or visual evidence to bolster significance.
> **Reply:**
> Thank you for your comment. Yes, our method is simple and effective, but it initially lacked a theoretical explanation. To address this, we have supplemented the theoretical analysis. The proposed method primarily enhances the diversity of the label space by incorporating multi-level synthetic image features into the existing text-based labels. Theoretically, we consider a more general scenario—how expanding multimodal labels improves OOD detection. We prove that, under certain conditions, the false positive rate ($\\text{FPR}\_\\lambda$) decreases as the number of multimodal labels increases, demonstrating that incorporating additional auxiliary modalities into labels enhances OOD detection performance. The specific theoretical details are as follows:
> ### Theoretical Analysis of Multimodal Label Enhancement
> **Core Contribution**
> We prove that expanding multimodal labels reduces OOD detection false positive rate ($\text{FPR}_\lambda$). The method enhances label diversity through multi-level synthetic image features, improving separability between ID/OOD samples.
>
> ---
> #### Key Theoretical Steps
> 1. ​**Multimodal Label Definition**
>    Define $N$-modal negative labels $ \\widetilde{Y}\_i =  \\{\widetilde{y}\_{i,1}, \\dots, \\widetilde{y}\_{i,N} \\}$, where $\\widetilde{y}\_{i,1}$ is the primary modality (text) and $\\widetilde{y}\_{i,2}, \\dots, \\widetilde{y}\_{i,N}$ are auxiliary modalities (synthetic image embeddings).
>
> 2. ​**Weight Allocation**
>    Assign non-uniform weights:
>    $$w\_j \= \\begin{cases} \\frac{a}{N}, & j=1 \\quad \\text{(primary modality weight)} \\\\ \\frac{1 \- \\frac{a}{N}}{N \- 1}, & j=2, \\dots, N \\quad \\text{(auxiliary modality weights)} \\end{cases}$$
>
> 3. ​**Aggregated Similarity Score**
>    Compute weighted similarity:
>    $$s_i = \frac{a}{N} s_{i,1} + \sum_{j=2}^N \frac{1 - \frac{a}{N}}{N - 1} s_{i,j}$$
>
> 4. ​**Statistical Properties**
>    For i.i.d. $s_{i,j} \sim (\mu, \sigma^2)$:
>    $$\mathbb{E}[s_i] = \mu, \quad \text{Var}(s_i) = \frac{\sigma^2}{N'(N)}, \quad N'(N) = \frac{N(N-1)}{a^2 + N - 2a}$$
>
> 5. ​**OOD Score Distribution**
>    Match count $c = \sum_{i=1}^M \mathbb{I}[s_i \geq \psi]$ follows:
>    $$c_{\text{in}} \sim \mathcal{N}(Mp_1, Mp_1(1-p_1)), \quad c_{\text{out}} \sim \mathcal{N}(Mp_2, Mp_2(1-p_2))$$
>    where $p_1 = 1-\Phi(k_1\sqrt{N'})$, $p_2 = 1-\Phi(k_2\sqrt{N'})$ with $k_1 = \frac{\psi-\mu_1}{\sigma}$, $k_2 = \frac{\psi-\mu_2}{\sigma}$.
>
> 6. ​**FPR Analysis**
>    $$\text{FPR}_\lambda = \Phi\left(\frac{\sqrt{M}(p_1 - p_2) + \sqrt{p_1(1-p_1)}\Phi^{-1}(\lambda)}{\sqrt{p_2(1-p_2)}}\right)$$
>
> 7. ​**Critical Result**
>    $\\frac{\partial \\text{FPR}\_\\lambda}{\\partial N} < 0 \\quad \\text{when} \\quad \\mu_1 + \\frac{\\sigma}{\\sqrt{N'}} > \\psi > \\mu_2$
>    ​**Conclusion**: Increasing $N$ strictly reduces $\text{FPR}_\lambda$.
>
> ---
>
> #### Remark on Assumptions
> - The i.i.d. assumption on $s_{i,j}$ can be relaxed to dependent variables.
> - Variance reduction ($\text{Var}(s_i) \propto 1/N$) remains the driving mechanism.
>
> **Theoretical Impact**: Provides mathematical guarantee for multimodal label enhancement in OOD detection. A detailed proof will be provided in the paper.
>
> **W2:** The main contribution of this paper is to enhance the zero-shot OOD detection capability of the CLIP model. Recently, there have been many related papers in this direction, such as \[1, 2\]. It is suggested that the authors add discussion and comparison. \[1\] LAPt: Label-driven Automated Prompt Tuning for OOD Detection with Vision-Language Models, ECCV2024 \[2\] CLIPScope: Enhancing Zero-Shot OOD Detection with Bayesian Scoring
> **Reply:**
> We appreciate the reviewer's valuable suggestion. In response, we have incorporated the LAPt method into the main experiments for comparison, as shown in Table A. Although LAPt adapts to ID data by fine-tuning prompt parameters, our method still achieves superior performance on the FPR95 metric.
>
>
> Table A: Experimental comparisons on the ImageNet-1k task, with iNaturalist, SUN, Places, and Texture used as out-of-distribution (OOD) data for evaluation.
>
> | Method | iNaturalist | SUN | Places | Texture | Average |
> | :---: | ----- | ----- | ----- | ----- | ----- |
> |  | FPR95 | FPR95 | FPR95 | FPR95 | FPR95 |
> | DualCnst(ours) | 1.29/99.65 | 17.60/95.89 | 31.91/92.13 | 42.15/90.51 | **23.24**/94.55 |
> | LAPt | 1.16/99.63 | 19.12/96.01 | 33.01/92.01 | 91.06/40.32 | 23.40/**94.68** |

---

### Official Review · Reviewer_ViPC · 2025-03-14

**Overall Recommendation:** 3

**Summary:**

This paper proposed a novel OOD approach named DualCnst, based on text-image dual consistency. In addition to detecting OOD samples by assessing the similarity between test images and ID/OOD label texts, this paper synthesizes OOD images using text-to-image models and incorporates the visual similarity between test images and ID/OOD images to enhance the model's performance. This paper achieves good performance on ImageNet-level OOD benchmarks.

**Claims And Evidence:**

Yes. This paper claims that incorporating visual information in VLM-based zero-shot settings is beneficial for OOD detection, as proven by experiments.

**Essential References Not Discussed:**

VLM-based OOD methods do take visual information into account. LoCoOp [1] aligns the semantic-relevant visual region with language features, and LSA [2] synthesizes high-likelihood ID features from class-specific Gaussian distributions to enhance the model's perception of ID semantics. Although these methods do not operate in a zero-shot setting, it is necessary to discuss and compare them, especially considering that this paper introduces an additional powerful yet time-consuming model like Stable-diffusion.
[1] LoCoOp: Few-Shot Out-of-Distribution Detection via Prompt Learning.
[2] Likelihood-Aware Semantic Alignment for Full-Spectrum Out-of-Distribution Detection.

**Experimental Designs Or Analyses:**

I have checked all experiments and analyses, and the specific issues are as follows:
1. I noticed in Appendix Sec. B.4 that for each OOD test dataset, the best-performing parameter \alpha was selected, which is very unreasonable. An important principle in OOD detection tasks is OOD agnosticism, meaning that in real-world environments, the categories and domain of OOD samples are broad and unknown. The use of OOD-specific hyperparameters in this paper conflicts with this principle.
2. I am confused about the fixed \omega in Sec.4 (such as in Line 313). According to the description of the method in this paper, the weights \omega of the intermediate and final layers of the image encoder should be different and sum up to 1. Why does a fixed \omega appear? I suspect that it might actually refer to \r?
3. The design of the Robust OOD Detection experiment in this paper is unreasonable. The experiment incorrectly replaces the ID dataset with a covariance shift ID (csID) dataset to explore the model's generalization on ID data. Instead, it should follow the full-spectrum OOD setup [1, 2], which retains the ID data while adding the csID data to the test samples. Otherwise, generalizing from ImageNet to ImageNet-R is too easy for models like CLIP and SD that have been exposed to large amounts of data, which fails to demonstrate the effectiveness of the proposed method.
[1] Full-Spectrum Out-of-Distribution Detection.
[2] OpenOOD v1.5: Enhanced Benchmark for Out-of-Distribution Detection.

**Methods And Evaluation Criteria:**

Although the novelty of the method in this paper is limited, it is reasonable within the field of OOD detection.

**Other Comments Or Suggestions:**

No more comments.

**Other Strengths And Weaknesses:**

Strengths: This paper is well-written in general, and the method design is reasonable.

Weaknesses: Aside from the issues in the experiments, the paper is limited in methodological novelty, as it combines NegLabel with a text-to-image approach. Furthermore, although the authors mention the computational burden advantages of this method compared to LMD, generating images with stable-diffusion remains a time-consuming process, which cannot meet the fast response requirements when deployed in real-world environments.

**Questions For Authors:**

No more questions.

**Relation To Broader Scientific Literature:**

This work introduces ID/OOD images generated by Stable-diffusion based on NegLabel [1], and uses visual similarity alongside text-image similarity from VLMs as an OOD scoring function. Additionally, prior works [2, 3] have implemented OOD detection using geneartion methods based on Stable-diffusion.
[1] Negative label guided ood detection with pretrained vision-language models.
[2] Unsupervised out-of-distribution detection with diffusion inpainting.
[3] Denoising diffusion models for out-of-distribution detection.

**Theoretical Claims:**

This paper does not provide significant theoretical proof, and no obvious errors were found in the formulas.

---

> ### Author Rebuttal · Authors · 2025-04-01
>
> Response to Reviewer ViPC
> We thank the reviewer ViPC for the valuable feedback. We addressed all the comments. Please find the point-to-point responses below. Any further comments and discussions are welcomed\!
> **W1:** The paper uses OOD-specific α (tuned per OOD dataset) , violating OOD agnosticism.
> **Reply:**
> We thank the reviewer for the feedback. **Experiments with a fixed α=0.1 achieve performance comparable to our original method while retaining significant advantages over baselines**. Due to space constraints, the complete table is available at:
> https://anonymous.4open.science/r/fjutlfy-31D8 Table A.
> Table A: Comparison with Other Baselines Using Fixed α=0.1.
>
> | Method | Average |
> | :---: | ----- |
> |  | FPR95 |
> | MCM | 42.74 |
> | NegLabel | 25.40 |
> | ours(Optimal α) | 23.05 |
> | ours(Fixed α=0. 1\) | 23.24 |
>
> **W2:** Sec4's fixed ω contradicts method's varying encoder ω (sum=1). Typo? (e.g., should be r)?
> **Reply:**
> Thank you for catching this \- we've corrected ω to r in the manuscript.
> **W3:** The current OOD detection framework oversimplifies evaluation. Adopting full-spectrum OOD would enable rigorous validation.
> **Reply:**
> Following OpenOOD v1.5 protocols, we now use ImageNet-1k/R as ID with near-OOD benchmarks (SSB-Hard, NINCO). Our method maintains superiority over baselines in this enhanced evaluation.
> Due to space constraints, the complete table is available at:
> https://anonymous.4open.science/r/fjutlfy-31D8 Table B.
> Table B: Robustness experiments on ImageNet-1k and ImageNet-R.
>
> | Method | Average |
> | :---: | :---- |
> |  | FPR95 |
> | MCM | 81.58 |
> | NegLabel | 50.04 |
> | DualCnst(ours) | **48.38** |
>
> **W4:** VLM-based OOD methods (e. g. , LoCoOp\[1\]/LSA\[2\]) already use visual semantics without costly generators (e. g. , SD) . Comparison needed to justify added complexity.
> **Reply:**
> We thank the reviewer for the suggestion. Added comparisons with LoCoOp (Table C) and LSA (Table D) show our method achieves superior performance over few-shot VLM baselines, with added compatibility for text enhancement (e.g., NegLabel). While LSA's official code is not fully released, we implemented it on their original dataset for fair evaluation. Due to space constraints, the complete table is available at:
> https://anonymous.4open.science/r/fjutlfy-31D8 Table C & D.
> Table C: Experimental comparisons on the ImageNet-1k benchmark.
>
> | Method | Average |
> | :---: | ----- |
> |  | FPR95 |
> | LoCoOpGL | 28.66 |
> | LoCoOpMCM | 33.98 |
> | DualCnst(ours) | **23.24** |
>
> Table D: Experimental comparisons on the ImageNet-1k task, with near and far datasets evaluated as OOD data.
>
> | Method | Average |
> | :---: | ----- |
> |  | FPR95 |
> | NegLabel | 51.60 |
> | LSA | 58.72 |
> | DualCnst(ours) | **50.03** |
>
> **W5: 1\.** High time cost of Stable Diffusion image generation undermines real-time deployment claims. **2\.** Limited methodological novelty: The approach primarily combines NegLabel with text-to-image synthesis
> **Reply:**
> **R1:**
> Regarding the efficiency of Stable Diffusion generation, we address the issue through two key optimizations:
> **First**, numerous accelerated versions of Stable Diffusion are now available. SDXL-Turbo achieves **10× faster generation** than SD1.5 (reducing time from 55m42s to 4m10s) while maintaining equivalent detection performance (Table E). This acceleration also enhances overall method performance (Table F).
> **Second, our synthetic images only need to be generated once, eliminating the need for repeated generation.** Due to space constraints, please refer to the full table at the link https://anonymous.4open.science/r/fjutlfy-31D8 Table E & F.
> Table E: Time comparison for accelerated SD models.
>
> | SD model | Time to generate ID images | Time to generate both ID and OOD images |
> | :---: | :---: | :---: |
> | SD1. 5 | 55m42s | 10h22m |
> | SDXL-Turbo | **4m10s** | **55m52s** |
>
> Table F: Performance comparison between SD models in the DualCnst method.
>
> | SD model | Average |
> | :---: | ----- |
> |  | FPR95 |
> | SD1. 5 | 23.24 |
> | SDXL-Turbo | **22.95** |
>
> **R2: ** To better understand our approach, we provide deeper theoretical insights into our approach, and have supplemented the theoretical analysis. The proposed method primarily enhances the diversity of the label space by incorporating multi-level synthetic image features into the existing text-based labels. Theoretically, **we consider a more general scenario—how expanding multimodal labels improves OOD detection. We prove that, under certain conditions, the false positive rate ($\\text{FPR}\_\\lambda$) decreases as the number of multimodal labels increases, demonstrating that incorporating more auxiliary modalities in labels enhances OOD detection performance.** Due to space limitations, the detailed theoretical analysis can be found in the response to Reviewer yyQ7-W1.

---

> > ### Comment · Reviewer_ViPC · 2025-04-08
> >
> > Thanks to the authors for the rebuttal. My concerns on the experimental part has been addressed in the rebuttal. But I still think this paper is limited in methodological novelty. Therefore I will increase my rating to WA.

---

> > > ### Author Response · Authors · 2025-04-09
> > >
> > > **Thanks for raising the score**
> > >
> > > Dear Reviewer ViPC,
> > >
> > > We thank the reviewer for raising the score! We sincerely appreciate your acknowledgment of our experimental revisions and your valuable input, which has helped strengthen our work. Regarding the novelty and contribution, we would like to clarify further:
> > > - **Simple yet Effective Framework (DualCnst):** Our approach uniquely integrates both semantic-textual similarity and visual similarity metrics between test samples and synthesized ID/OOD labels, significantly improving VLM-based OOD detection accuracy.
> > > - **Theoretical Analysis:** As detailed in our supplemental materials, we theoretically demonstrate that leveraging multimodal label spaces (text + synthetic images) reduces the false positive rate under certain conditions. This proves that incorporating auxiliary modalities enhances OOD detection performance.
> > >
> > > Best regards,
> > >
> > > Authors of 3561

---

### Decision · Program_Chairs · 2025-05-01

**Decision:**

Reject

**Comment:**

This paper receives three Weak Accepts and one Weak Reject. In the initial review round, reviewers acknowledged the rationality and simplicity of the method (ViPC, yyQ7), the strong OOD detection performance of the method (yyQ7, Ci5u, aTAh), and the clarity of the paper (ViPC, Ci5u). However, concerns were also raised about the novelty of the method (ViPC, yyQ7, Ci5u), lack of theoretical basis (yyQ7, aTAh), computational cost (ViPC, Ci5u, aTAh), and lack of comparisons and analysis (yyQ7, Ci5u, aTAh).

The authors submitted a rebuttal. All reviewers actively discussed the problems during the Author-Reviewer Discussion and/or AC-Reviewer Discussion periods. Indeed, the authors did a great job of addressing the reviewers' concerns by providing an additional theoretical proof and experimental results, which successfully resolved some of the reviewers concerns. However, all reviewers thought that the novelty is still not strong enough. Regarding computation time, the effect of SDXL-Turbo is impressive but can still be a major problem: as long as the set of ID categories is fixed in advance, it is possible to obtain OOD labels and generate the pseudo OOD images offline. However, if the set of ID categories cannot be fixed in advance, for example, online generation is required, which may require 56 mins for classification at least one time. Consequently, Reviewer Ci5u, who had initially proposed weak accept, agreed with these weaknesses during the AC-Reviewer Discussion and expressed a leaning toward rejection.